# The impact of traffic on air quality in Ireland: insights from the simultaneous kerbside and sub-urban monitoring of submicron aerosols

Chunshui Lin[1,2,3], Darius Ceburnis[1], Wei Xu[1,2], Eimear Heffernan[4], Stig Hellebust[4], John Gallagher[5], Ru-Jin Huang[1,2,3*], Colin O'Dowd[1*], and Jurgita Ovadnevaite[1]

[1]School of Physics, Ryan Institute's Centre for Climate and Air Pollution Studies, National University of Ireland Galway. University Road, Galway. H91 CF50, Ireland
[2]State Key Laboratory of Loess and Quaternary Geology and Key Laboratory of Aerosol Chemistry and Physics, Institute of Earth Environment, Chinese Academy of Sciences, 710061, Xi'an, China
[3]Center for Excellence in Quaternary Science and Global Change, Chinese Academy of Sciences, Xi'an 710061, China
[4]School of Chemistry and Environmental Research Institute, University College Cork, Cork, Ireland
[5]Department of Civil, Structural & Environmental Engineering, Trinity College Dublin, the University of Dublin, Ireland.

*Correspondence to*: Ru-Jin Huang (rujin.huang@ieecas.cn) and Colin O'Dowd (colin.odowd@nuigalway.ie)

**Abstract:** To evaluate the impact of traffic on urban air quality, the chemical composition of submicron aerosols ($PM_1$) and sources of organic aerosol (OA) were simultaneously investigated at a kerbside site in Dublin city centre and a residential site in sub-urban Dublin (~5 km apart) from 4 September to 9 November in 2018. Through the detailed comparison of one-week non-heating period from 10 to 17 September and one-week heating period from 27 October to 4 November, black carbon (BC) was found to be the most dominant component (38-55% or 5.6-7.1 µg m$^{-3}$) of $PM_1$ at the kerbside while OA was the most important (46-64% of $PM_1$ or 1.0-8.1 µg m$^{-3}$) at the residential site. The daily and weekly cycle of BC at the kerbside during the non-heating period pointed to the major source of vehicular emissions, consistent with that for nitrogen oxides ($NO_x$). However, traffic emissions were found to have a minor impact on air quality at the residential site due to its distance from traffic sources, as well as the effects of wind speed and wind direction. As a result of vehicular emissions and the street canyon effect, the kerbside increment (from urban background) ratio of up to 25:1 was found for BC during the non-heating period but reduced to 10:1 during the heating period due to the additional sources of solid fuel burning impacting the air quality at both sites simultaneously. OA source analysis shows only 16-28% (0.9-1.0 µg m$^{-3}$; upper limit for traffic due to the additional heating source of hydrocarbon-like OA (HOA)) of OA at the kerbside associated with vehicular emissions, with higher contributions from cooking (18-36% or 1.2 µg m$^{-3}$), solid fuel burning (38% or 2.4 µg m$^{-3}$, resolved only during the heating period), and oxygenated OA (29-37% or 1.2-1.9 µg m$^{-3}$). At the residential site, solid fuel burning contributed to 60% (4.9 µg m$^{-3}$) of OA during the heating period, while oxygenated OA accounted for almost 65% (0.6 µg m$^{-3}$) of OA during the non-heating period. Based on simultaneous investigation of $PM_1$ at different urban settings (i.e. residential vs kerbside), this study highlights temporal and spatial variability of sources within Dublin city centre and the need for additional aerosol characterization studies to improve targeted mitigation solutions for greater impact on urban air quality. Moreover, traffic and

residential heating may hold different implications for health and climate as indicated by the significant increment of BC at the kerbside and the large geographic impact of OA from residential heating at both the kerbside and residential sites.

## 1    Introduction

Aerosol particles adversely affect human health, causing morbidity and premature mortality (Pope III et al., 2002; Cohen et al., 2017; Burnett et al., 2018). In particular, urban areas are usually more polluted than rural areas due to the proximity to pollution sources including traffic, cooking, and various other activities such as residential heating. To make things worse, the street canyon effects in urban areas result in poor dispersion conditions, trapping the pollutants in downtown areas where the streets are flanked by buildings on both sides creating a canyon-like environment (Fuzzi et al., 2015). The term 'urban increment' is defined as the increase in air pollution parameters in the urban background above the rural surroundings while 'kerbside increment' is defined as the increase in concentrations at a kerbside or street site relative to the urban background (Lenschow et al., 2001; Mues et al., 2013; Fuzzi et al., 2015). Models often underestimate the urban increment let alone kerbside increment, due to the poor grid resolution and not taking into account the urban micro-environment such as the urban canyon and urban heat-island effects (Mues et al., 2013). Therefore, field campaigns, especially those simultaneously measuring at different settings (e.g., residential site vs. kerbside) in the urban area, are required to study the spatial variation of aerosols and their sources as well as to provide constraints for the models.

Aerosol can be broadly categorized into two classes: primary and secondary (Fuzzi et al., 2015). Primary aerosols are directly emitted from their emissions sources such as traffic, cooking, and biomass burning while secondary aerosols are formed from their corresponding precursor gases (An et al., 2019). Primary aerosols often show enhanced concentrations when close to their sources (e.g., at the kerbside), while secondary aerosols are more homogeneously distributed (Hallquist et al., 2009; Zhang et al., 2015; Gentner et al., 2017). Mohr et al. (2011) showed a kerbside increment of up to 11 μg m$^{-3}$ for black carbon (BC) and 2.5 μg m$^{-3}$ for hydrocarbon-like organic aerosol (HOA) at a roadside in Zurich with the mobile measurements using an Aerosol Mass Spectrometer (AMS). While mobile or on-road measurements can provide a detailed characterization of the spatial variation of the chemical components and sources (e.g., traffic) of aerosols, mobile measurements can also be strongly influenced by the emission from a specific source e.g., from an old type truck (or high emitters), not representing the average fleet (Fuzzi et al., 2015). Moreover, mobile measurements are usually conducted at a specific time of the day, failing to capture the temporal variation of aerosols over longer periods (e.g., days to months) (Mohr et al., 2011; Fuzzi et al., 2015). Fixed installation of instruments at multiple sites simultaneously can, therefore, provide more detailed information about both the temporal and spatial variation of aerosols. Crippa et al. (2013) investigated the temporal evolution of the submicron aerosol at three sites (one urban centre and two urban background sites) in Paris during wintertime. In contrast to the finding by Mohr et al. (2011), Crippa et al. (2013) concluded that the submicron aerosol in Paris was dominated by regional transport and that the emissions from Paris itself had an insignificant impact on the urban backgrounds. The discrepancies between the individual studies may be related to the specific geographic environment and the periods of the measurement (Fuzzi et al., 2015).

Dublin, the capital city of the Republic of Ireland in Western Europe, is home to ~1 million people. Previous study conducted at a suburban site in the residential area in Dublin has shown that air quality was strongly influenced by residential heating sources in winter (Lin et al., 2018). However, the impact of traffic emissions on this residential site was shown to be minor probably due to the distance (~500 m) from the nearest roads, as well as the strict emission standard (Lin et al., 2018). The minor influence of traffic at the residential site was also shown in a west coast city of Galway in Ireland during both summer (Lin et al., 2019a) and winter conditions (Lin et al., 2017). In particular, the diurnal pattern of HOA shows largely enhanced concentration in the evening when compared to that during the morning rush hours, suggesting residential heating is the major source of HOA in suburban Dublin as shown in our previous studies (Lin et al., 2018; Lin et al., 2019b). The heating source of HOA is also reported at the urban background site in Paris in addition to traffic (Petit et al., 2014; Zhang et al., 2019). However, the relative importance of traffic and heating to HOA in different urban settings (e.g., kerbside and residential) and different seasons (e.g., heating and non-heating) remain poorly understood.

In this study, the chemical composition of submicron aerosol (PM$_1$) at both kerbside and residential sites was simultaneously measured from 4 September to 9 November 2018 using an Aerodyne Aerosol Chemical Speciation Monitor (ACSM) and an Aethalometer. The chemical composition of PM$_1$ and sources of OA at these two sites were explicitly characterized and compared during both the non-heating period and the heating period. Finally, the spatial variation and kerbside increment of PM$_1$, as well as the implications for pollution mitigation strategies in Ireland were discussed.

## 2    Experimental

### 2.1    Sampling sites

Chemical compositions of PM$_1$ were simultaneously measured at a kerbside site in Dublin city centre and a residential site in suburban Dublin, at a distance of ~5 km from the kerbside site (see the map in Fig. S1) while PM$_{2.5}$ was sampled at Rathmines (https://www.epa.ie/air/quality/data/rm/), another residential site at a distance of ~3 km from the kerbside and residential sites. The kerbside site is adjacent to a heavily trafficked street (i.e., Pearse Street) with a traffic flow of ~46,000 vehicles per day, 76% of which consist of private cars, 13% of light goods vehicles, 7% of heavy goods vehicles, 2% of buses, and 2% of motorcycles (Fu et al., 2017). Pearse Street is characterised as an almost symmetrical street canyon i.e. rows of buildings on both sides of the road that are of equal height to the width of the street, and this affects local dispersion of air pollutants (Gallagher et al., 2013; Gallagher, 2016). It is worth noting that a bus stop was positioned ~20 m south from the sampling inlet. Measurements of submicron aerosol at the kerbside took place at ~ 3 m from the busy street. The residential site is located on the campus of University College Dublin (UCD) in south Dublin. The nearest road is ~500 m away from the sampling site at UCD. Measurements at the site were conducted on the roof of the Science building (~30 m above the ground) at UCD from 4 September to 9 November 2018.

## 2.2 Instrumentation

A quadrupole Aerosol Chemical Speciation Monitor (Q-ACSM, Aerodyne Research Inc.) and an aethalometer (AE-33, from Magee Scientific) were deployed at the kerbside site to measure the non-refractory $PM_1$ (NR-$PM_1$) component (i.e. OA, sulfate, nitrate, ammonium, and chloride) and BC, respectively. At the residential site, another Q-ACSM and aethalometer (AE-16) were deployed to measure the $PM_1$ (NR-$PM_1$ and BC) composition simultaneously. At each site, ACSM and AE-33/16 were sampling from the same $PM_{2.5}$ inlet line with isokinetic flow splitting. A detailed description of the ACSM and its operation is given by Ng et al. (2011b) and in our previous study (Lin et al., 2018). Briefly, an ACSM consists of a particle sampling inlet, three vacuum chambers and a quadrupole mass spectrometer. During sampling, the ambient air was drawn into the cyclone with a size cut-off of 2.5 μm at a flow rate of 3 L $min^{-1}$ to remove coarse particles. A Nafion dryer was used to dry the ambient air before reaching the ACSM inlet. In the ACSM particle sampling inlet, the dried aerosol particles were focused into a narrow aerosol beam through the aerodynamic lens system. The beam was directed onto a hot tungsten oven (~600 °C) where the particles were vaporized. The vaporized molecules were ionized by electron impact (70 eV) and the resulting ions were analyzed by a quadrupole mass spectrometer. Both ACSMs were calibrated with ammonium nitrate and ammonium sulfate following the procedure described by Ng et al. (2011b). Briefly, monodispersed 300 nm ammonium nitrate/ammonium sulfate particles were generated by an atomizer (Model 9302, TSI) and size-selected by a differential mobility analyzer (DMA; TSI model 3080), and subsequently introduced into the Q-ACSMs to determine the response factor (RF), as well as the relative ionization efficiencies (RIEs) of ammonium and sulfate.

The time resolution of ACSM at the kerbside was set at 5 minutes to capture the faster changing variation of PM composition adjacent to the busy road while a 1-hour interval was used at the residential site to reduce uncertainty in measurement due to the relatively low concentrations for most of the time. The ACSM standard data analysis software (v 1.6.1.0) in Igor 6.37 (WaveMetrics Inc.) was used to process the mass concentrations of NR-$PM_1$ components. To account for the aerosol losses during sampling (Middlebrook et al., 2012), a composition dependent collection efficiency (CDCE) was applied. The CDCE (averaged at ~0.5) corrected NR-$PM_1$ shows a good agreement with the $PM_{2.5}$ measurement at another residential site in suburban Dublin (i.e., Rathmines, Fig. S1) where a tapered element oscillating microbalance (ThermoFisher Scientific) was deployed with 1 h resolution.

The AE-33 was deployed to measure black carbon (BC) at the kerbside with a time resolution of 1 minute while AE-16 at the residential site had a time resolution of 5 minutes. AE-33 measures light absorption at seven wavelengths (370, 470, 520, 590, 660, 880, and 950 nm) (Drinovec et al., 2015) while AE-16 measures light absorption solely at 880 nm. BC mass concentration was calculated from the change in optical attenuation at 880 nm in the selected time interval using the mass absorption cross-section 7.77 $m^2$ $g^{-1}$ (Drinovec et al., 2015).

Carbon monoxide analyzer (Thermo Scientific Model 48i) was employed to measure the CO mixing ratios with a time resolution of 1 minute at the kerbside. NO-$NO_2$-$NO_x$ analyzer (Thermo Scientific Model 42i) was employed to sample $NO_x$ with a time resolution of 1 minute at the kerbside and residential site simultaneously. Meteorological variables (temperature,

relative humidity, wind speed, and wind direction) with a time resolution of 1 hour were recorded at the meteorological stations
(Irish meteorological service) of Dublin airport (~5 km north to the kerbside sampling site).

**2.3    OA Source apportionment**

Positive matrix factorization (PMF) was utilized for OA source apportionment on the interface of SoFi 6.A1 (Canonaco et al.,
2013). Firstly, unconstrained (or free) PMF was applied to examine the OA factors based on their mass spectral profiles,
diurnal patterns, and correlations with external measurements (e.g., NOx and BC). However,  the free PMF solution could
potentially lead to solutions with mixed factors or inaccurate factor attributions when some factors have similar temporal
variation and/or factor profiles (e.g., COA and HOA) (Canonaco et al., 2013). This is especially true for the heating-related
OA factors during the heating-season in Ireland where oil, peat, wood, and coal were commonly used as domestic heating fuels
according to Central Statistics Office (CSO, 2016) and Sustainable Energy Authority of Ireland (SEAI, 2018). Moreover,
previous offline (i.e., filter-based) and online (i.e., ACSM/AMS) studies in Ireland winter have identified ambient OA being
associated with the residential burning of oil, peat, coal, and wood burning (Kourtchev et al., 2011; Dall'Osto et al., 2013; Lin
et al., 2018). In this study, free PMF resulted in highly mixed OA factors (See the supplementary section for the free PMF
solutions). Thus, based on this priori information on emission sources, PMF with the multilinear engine (ME-2) was applied
to get more environmentally meaningful solutions by constraining the reference profiles with $a$ value approach (Canonaco et
al., 2013; Lin et al., 2017).

   The reference profiles of peat, coal, and wood were taken from a previous study in which the mass spectral signatures of
peat, coal, and wood burning OA were characterized by simulating their burning in a typical Irish stove with an ACSM (Lin
et al., 2017). HOA and COA were obtained from a study in Paris by Crippa et al. (2013). A sensitivity analysis was undertaken
by varying the $a$ values (0-0.5 or 0-50% variation) to evaluate the OA factor contribution at different levels of constraint on
the reference factor. Moreover, a bootstrap-based resampling strategy with a total of 500 runs was applied to examine the
statistical uncertainty (https://datalystica.com/sofi). The following criteria are used for the selection from these ME-2 runs to
get the most optimized solution:

   1. Correlation between the time series of HOA and NOx or BC in the morning hours (7:00-10:00) during weekdays. HOA
and NOx ($R^2$>0.4)/BC ($R^2$>0.6) have a common source from traffic emissions in these morning hours.
2. Diurnal cycle of COA. The COA concentrations during mealtime should be higher. In this study, the ratio of COA at 13:00
to the average of COA of 10 and 11 am was monitored.

   3. Multi-linear regression between BC and heating-related factors (i.e., HOA (or oil burning), peat, wood, and coal burning)
in the evening (19:00-23:00), assuming only the heating-related factors contributed to BC in the evening.

   4. Fraction of m/z 60 (i.e., *f60*) for peat and wood burning OA factors should be higher than 0.006 (Cubison et al., 2011; Lin
et al., 2017).

   5. Fraction of m/z 44 (i.e., *f44*) for the unconstrained factor (i.e., OOA). OOA should have higher *f44* than primary factors.

By inspecting the defined criteria, the ME-2 runs that meet these criteria were selected and averaged as the most optimized solution.

## 3 Results and discussion

### 3.1 Overview of aerosol measurement

Figure 1 shows the time series of submicron organic aerosol (OA), sulfate, nitrate, ammonium, chloride, and black carbon (BC) at both the kerbside and the residential site, as well as the time series of $PM_{2.5}$ at Rathmines from 4 September to 9 November 2018. The air quality during the early stage of the sampling period (before 30 September) showed limited influence from residential heating emissions as observed by few pollution spikes in the evening when compared to a later period with prominent evening spikes observed at all three sites (From 1 October to 09 November; Fig. 1). The three sampling sites (i.e., kerbside, residential, and Rathmines) are within a 5 km radius in Dublin city and the time series of $PM_1$ and $PM_{2.5}$ were well correlated with linear correlation coefficients determination ($R^2$) in the range of 0.56-0.83 and slopes of 0.72-0.88 (Fig. S1). The good time series correlation between the three sites was mainly driven by the pollution events during the heating period while the slopes of 0.72-0.88 suggested $PM_1$ explained, on average, 72-88% of $PM_{2.5}$ mass. The poorer correlation ($R^2$ of 0.56; Fig. S1) between the kerbside $PM_1$ and Rathmines $PM_{2.5}$ than between the residential $PM_1$ and Rathmines $PM_{2.5}$ ($R^2$ of 0.83) was due to traffic emissions which had a greater impact on the kerbside than at both the residential site and Rathmines.

Over the entire period, the mean mass concentration of $PM_1$ was 11.7 $\pm$ 9.7 (one standard deviation) µg m$^{-3}$ at the kerbside with black carbon accounting for 50% of the total $PM_1$ mass (Fig. 1a), followed by OA (31%), nitrate (6%), sulfate (5%), ammonium (5%), and chloride (2%). At the residential site, the mean $PM_1$ concentration (6.6 $\pm$ 3.1 µg m$^{-3}$) was roughly half of that at the Kerbside (Fig. 1b). However, the chemical compositions of $PM_1$ at the residential site was dominated by OA (59% of $PM_1$), followed by BC (13%). The total inorganic aerosols (sum of sulfate, nitrate, ammonium, and chloride) accounted for 28% of $PM_1$ at the residential site, slightly higher than at the kerbside (18%), though the concentrations were comparable (1.8 vs. 2.1 µg m$^{-3}$) between the two sites, suggesting regional sources/formations rather than traffic emissions. In contrast, the greater abundance of BC at the kerbside suggests traffic emissions comprised a high fraction of BC which, however, had a limited impact on the residential site. In particular, BC at the kerbside showed two rush hour peaks, confirming traffic was its major source (Fig. S2). However, the BC was up to 25 times (7 times on average) lower at the residential site during the rush hours (Fig. S2), suggesting a minor impact of traffic on the air quality at the residential site due to the effects of the wind direction and the distance from the road. In addition to the rush hour peaks of BC, an additional BC peak in the evening was also observed at both sites due to the emissions from domestic heating activities (discussed later).

To better understand the evolution of chemical composition and concentration of $PM_1$, two periods, with one week of the non-heating period from 10 to 17 September and one week of the heating period from 27 October to 4 November, were selected for a more detailed comparison (Fig. 2). During non-heating, BC showed the largest kerbside increment, with an increment ratio of 15 (median value) (Fig. 3). During heating, the BC increment ratio was slightly lower (10) primarily due to the

additional emission sources of solid fuel burning affecting both sites simultaneously. The kerbside increment of BC was primarily due to vehicular emissions and this was corroborated by the gas pollutant of $NO_x$ which showed an increment ratio of 7-11 (Fig. 3). Note that our measurement represents the average fleet in Dublin and thus the high kerbside increment for BC in Dublin has significant implications for the potentially higher exposure risk at the kerbside. Compared to BC, the kerbside increment of OA was less significant because traffic was not a major source of OA (discussed in Sect. 3.4). However, a higher OA increment ratio (3; median value) was also seen during non-heating than during heating (~2).

The meteorological parameters including wind speed, wind direction, relative humidity (RH), and temperature during the non-heating and heating period are shown in Fig. S3. Specifically, during the non-heating period, south-westerly winds were prevailing, with an average wind speed of 4.5 m s$^{-1}$, ranging from 1.5 to 9.0 m s$^{-1}$ while, during the heating period, the wind speeds were slightly lower with an average of 3.7 m s$^{-1}$ (range 0.5-8.5 m s$^{-1}$). The low wind speeds were accompanied by the northerly and easterly winds during the heating period. The temperature during non-heating averaged 13.1 °C, ranging from 8.6 °C to 18.2 °C while it was lower during heating with an average of 4.5 °C (range -3.9-11.6 °C). However, the RH during non-heating and heating were similar, with an average RH of 78.7% (range 57-96%) for non-heating and 78.9% (range 44-98%) for heating.

## 3.2 Mass concentration and chemical composition of PM$_1$ during non-heating

### 3.2.1 PM$_1$ at the kerbside during non-heating

Figure 2a shows the time series zoomed in for the non-heating and heating periods and their relative fractions of PM$_1$ component (i.e., OA, sulfate, nitrate, ammonium, and chloride, and BC). BC was the most dominant PM$_1$ component (Table 1), on average accounting for over half (55% or 5.6 μg m$^{-3}$) of PM$_1$, followed by OA (32% or 3.3 μg m$^{-3}$), sulfate (5% or 0.5 μg m$^{-3}$), ammonium (4% or 0.4 μg m$^{-3}$), nitrate (3% or 0.3 μg m$^{-3}$), and chloride (1% or 0.1 μg m$^{-3}$). In particular, BC spikes (> 15 μg m$^{-3}$; top 5 percentiles) affected the local air quality substantially during the daytime (6:00 – 21:00, local time, Fig. 4a) and showed higher intensity during rush hours on weekdays. As a result, the averaged BC diurnal profile showed both morning (9:00) and afternoon (17:00) rush-hour peaks (>10 μg m$^{-3}$; Fig. 4a). Similar patterns were also observed for $NO_x$ (Fig. S4) and CO (Fig. S5) at the kerbside, confirming common traffic sources. Specifically, both NOx and CO showed rush hour peaks during weekdays while such rush hour peaks during weekends were not as prominent as during weekends, consistent with the traffic pattern in Dublin (Fu et al., 2017). The average mixing ratios for NOx and CO were 54.1 ppb (in the range of 5.4-200.0 ppb) and 0.2 ppm (in the range of 0.05-0.8 ppm), respectively.

BC spikes were on the time scale of minutes, indicating certain types of vehicles were the major cause of such pollution plumes. The diesel-powered public buses were firstly suspected as the major emitters of BC because a bus stop was nearby, about ~20 m away from the sampling site. Note the bus services usually start at ~5:30 and end at ~23:00 in downtown Dublin. However, BC spikes were also observed during other times e.g., from 23:00 to 5:30 when public bus services were not available.

Moreover, higher intensity and frequency of BC spikes were observed during the morning and evening rush hour peaks. Thus, in addition to the buses, other types of vehicles (e.g., private cars) were also potential contributors to air pollution. Specifically, ~50 vehicles (manual count) were jammed along the nearby street during rush hours and higher emissions might be associated with the cold starts and idling speeds during such traffic jams. Additionally, the street canyon effect, making the pollutants hard to disperse, were also important factors in driving the high BC concentrations at the kerbside. In particular, the street canyon effect was evidenced by the high background BC concentration (3.0 $\mu g\ m^{-3}$) during the night (from 23:00 to 5:00) when traffic flow was at the lowest (Fig. 4a). As a comparison, the average BC was only 0.4 $\mu g\ m^{-3}$ at the residential site during the non-heating period (Table 1).

Similar to BC at the kerbside, OA also shows spikes with concentrations of >5 $\mu g\ m^{-3}$ (top 5 percentiles) from 6:00 to 21:00 (Fig. 4a). However, in addition to traffic, cooking and secondary formation were also contributing to the OA mass at the kerbside (discussed in Sect. 3.4). In contrast to BC and OA, the time series of the measured inorganic aerosols were relatively smooth (Fig. 2a), and did not exhibit any obvious influence from traffic. This is not surprising because vehicles are not direct emitters of these inorganic species, and the introduction of the threshold of 10 ppm (by mass) sulfur for gas/diesel oil led to very low sulphur emissions in Dublin (Regulations, 2008). On average, the sum of inorganic aerosol accounted for a relatively small fraction (13%) of the total $PM_1$ (Fig. 2a).

### 3.2.2 $PM_1$ at the residential site during non-heating

The time series of OA, sulfate, nitrate, ammonium, and chloride at the residential site during non-heating are shown in Fig. 2b. Over this period, the average concentration of $PM_1$ was 2.3 $\mu g\ m^{-3}$ at the residential site, around five times lower than that at the kerbside (Table 1). OA was the most dominant species at the residential site, accounting for 46% (1.0 $\mu g\ m^{-3}$) of $PM_1$ (Fig. 2b), followed by BC (17% or 0.4 $\mu g\ m^{-3}$), sulfate (17% or 0.4 $\mu g\ m^{-3}$), ammonium (14% or 0.3 $\mu g\ m^{-3}$), nitrate (5% or 0.1 $\mu g\ m^{-3}$), and chloride (1% or <0.1 $\mu g\ m^{-3}$). Although the measurements at the residential and kerbside were conducted simultaneously, all of the $PM_1$ components showed no obvious trend that could be associated with vehicular emissions at the residential site (Fig. 4b), indicating the impact of traffic emissions from the kerbside had a minor impact on the residential site. Consistently, $NO_x$ concentrations were very low with an average mixing ratio of $NO_x$ of 2.6 ppb (median was 2.0 ppb; Table 1). As discussed above, the low impact of vehicular emissions on air quality at the residential site was associated with the distance from the emission sources, as well as the effects of wind speed and wind direction. Specifically, the distance between the residential site and the nearest road is ~500 m and ~5 km away from the city centre. The south-westerly wind was dominant during the non-heating period with an average wind speed of 4.5 $m\ s^{-1}$, ranging from 1.5 to 9.0 $m\ s^{-1}$ (Fig. S3).

### 3.3 Mass concentration and chemical composition of PM$_1$ during heating

#### 3.3.1 PM$_1$ at the kerbside during heating

During the heating period, large pollution spikes were observed in the evening (20:00-23:00) with a simultaneous increase in all NR-PM$_1$ components and BC (Fig. 2c and 4c). Specifically, in the evening on 31 October, the peak PM$_1$ concentration was 134.0 µg m$^{-3}$ (Fig. 2c) which was more than double of the rush hour peak on the same day. Moreover, the evening pollution expanded over the entire evening and went into the night (Fig. 4c), demonstrating the extensive impact of residential heating activities. On the same day, BC concentration increased up to 37.5 µg m$^{-3}$ in the evening, over five times higher than the average (7.1 µg m$^{-3}$). The simultaneous increase in all NR-PM$_1$ components along with BC suggests common heating sources in the evening. Note that the shallower boundary layer in the evening was also partly contributing to the elevated concentrations of the PM$_1$ components.

Over the entire heating period, the average PM$_1$ concentration was 18.4 µg m$^{-3}$, approximately two times higher than that during non-heating (Table 1). While the average BC concentration increased from 5.6 µg m$^{-3}$ during non-heating to 7.1 µg m$^{-3}$ during heating, the corresponding BC fraction decreased from 55% to 38%. This is due to the relatively high emissions of other PM$_1$ components (e.g., OA) which were associated with heating sources. Specifically, the average OA concentration was doubled during heating (6.5 µg m$^{-3}$) than during non-heating (3.3 µg m$^{-3}$) with the corresponding fraction increasing from 32% to 35%. Moreover, the average concentration and fraction of sulfate, nitrate, ammonium, and chloride also increased. On average, the fraction of inorganic components increased from 13% (1.3 µg m$^{-3}$) during non-heating to 27% (6.1 µg m$^{-3}$) during heating. In particular, nitrate saw a large increase from 3% (0.3 µg m$^{-3}$) during non-heating to 9% (1.6 µg m$^{-3}$) during heating, partly due to the cold temperature (Fig. S4) which favoured the gas-to-particle partitioning of semi-volatile NH$_4$NO$_3$. Moreover, the maximum chloride concentration (28.5 µg m$^{-3}$; Fig. 2c) was observed in the evening of 31 October along with other species, suggesting emission sources from solid fuel burning (Lin et al., 2017). While the diurnal pattern of PM$_1$ during the heating period still showed two rush hour peaks as found during the non-heating period (Fig. 4c), the higher PM$_1$ peak in the evening again highlighted the importance of the heating emissions.

#### 3.3.2 PM$_1$ at the residential site during heating

The time series of the measured PM$_1$ components at the residential site during heating is shown in Fig. 2d. During this period, the pollution spikes with a simultaneous increase in the concentrations of NR-PM$_1$ and BC were observed almost every evening. This was in great contrast to that during non-heating when no clear pattern in NR-PM$_1$ species was observed (Fig. 4b and 4d). As discussed above, the pollution spikes of NR-PM$_1$ at the residential site showed a simultaneous increase with that at the kerbside (Fig. 2c) and the PM$_{2.5}$ at the Rathmines station (Fig. 1c), indicating the three sites were affected by similar residential burning sources and air masses. In particular, the maximum concentration of chloride (18.3 µg m$^{-3}$), observed in the evening of 31 October at the residential site, suggested emissions from solid fuel burning as found at the kerbside.

Over the entire heating period, the average PM$_1$ concentration was 12.7 μg m$^{-3}$, which was six times higher than during non-heating. On average, OA accounted for 63% (8.1 μg m$^{-3}$) of PM$_1$, making it the most dominant component (Fig. 2d), followed by BC (9% or 2.4 μg m$^{-3}$), nitrate (8% or 1.1 μg m$^{-3}$), ammonium (8% or 1.0 μg m$^{-3}$), sulfate (7% or 0.9 μg m$^{-3}$), and chloride (4% or 0.6 μg m$^{-3}$). Similar to that during the non-heating period, the averaged diurnal profile of the PM$_1$ components during the heating period showed a weak impact from traffic emission during the day (Fig. 4d). In contrast, the large increase of PM$_1$ components in the evening and night indicated substantial contribution from the heating sources.

## 3.4    Sources of OA

### 3.4.1    OA factors at the kerbside during non-heating

To resolve the OA sources, free PMF was firstly conducted on the OA matrix. The solution that best represented the data was the three-factor solution because the solutions with more factors provided no meaningful but splitting from the already resolved factors (see Fig. S6 and more details in the supplementary). The three-factor solution identified a hydrocarbon-like OA (HOA) factor from traffic, cooking-like OA (COA) factor from cooking sources, and an oxygenated OA (OOA) factor corresponding to the secondary processes. However, free PMF provided only a sub-optimal solution as the profile of COA contained no m/z 43 which underestimated its contribution or over-estimated the contributions from other factors (Fig. S6). Therefore, ME-2 (Canonaco et al., 2013) was utilized to constrain the reference profiles of HOA and COA (Crippa et al., 2013) with 0-50% constraints (i.e., *a* value of 0-0.5; see Sect. 2.3) to better evaluate their contributions.

Figure 5 shows the mass spectra, diurnal pattern, and relative contribution of HOA, COA, and OOA of the optimal ME-2 solution during non-heating at the kerbside.  The HOA profile was dominated by peaks at *m/z* of 27, 29, 41, 43, 55, and 57, characteristic of the hydrocarbon ion series of $[C_nH_{2n+1}]^+$ and $[C_nH_{2n-1}]^+$. The diurnal cycle of HOA featured two rush hour peaks which were consistent with that of BC (Fig. 4a), confirming its traffic source. During the early morning (from 00:00 to 5:00), the HOA concentration was very low (0.1 μg m$^{-3}$), indicating a low background HOA level. From 6:00, HOA started to increase with a peak HOA value of 1.5 μg m$^{-3}$ at 8:00. After the morning rush hour peak, the HOA concentration was constantly high (1 μg m$^{-3}$) and reached 1.3 μg m$^{-3}$ at the afternoon rush hour peak (17:00). The HOA returned to the background level at around 23:00, corresponding to the reduced traffic flow during the night (Fu et al., 2017). Despite being adjacent to the busy road in Dublin city centre, on average, HOA accounted for 27.8 % (0.9 μg m$^{-3}$) of the total OA (Fig. 5c) with rest attributing to COA (35.7% or 1.2 μg m$^{-3}$) and OOA (36.6% or 1.2 μg m$^{-3}$). During the morning rush hours from 7 am to 10 am, HOA increased its shares to 49.1% of the total OA (Fig. 5d), suggesting a more important role of HOA during rush hours.

The COA profile is characterized with a *f55* to *f57* ratio of 2.6 (where *f55* and *f57* are the fraction of *m/z* 55 and 57 to the total organic mass, respectively) which was higher than that for HOA (0.9) but was in the range of 2.2-2.8 typical found for COA (Mohr et al., 2012). The identification of COA was associated with the location of the sampling site in Dublin city centre with some restaurants around. The diurnal pattern of COA showed a lunchtime peak at 13:00 and a dinner time peak at 20:00,

corresponding to the mealtimes. Higher dinner time COA peak (2.7 µg m$^{-3}$) was observed than lunchtime peak (1.6 µg m$^{-3}$) likely due to higher emissions during the evening coupled with relatively low evening temperatures (Fig. S3) and shallower boundary layer. Similar diurnal patterns for COA were also observed in Barcelona (Mohr et al., 2012), Paris (Crippa et al., 2013), London and Manchester (Allan et al., 2010). On average, COA accounted for 35.7% of OA, but over the evening hours from 19:00 to 22:00, COA increased its fraction to 51.4% of OA.

The OOA profile (Fig. 5) resembles the low-volatility OOA (LV-OOA; R$^2$ of 0.94 between the OOA and the reference LV-OOA profile) (Ng et al., 2011a) which typically represents well-aged SOA and correlates better with non-volatile secondary species such as sulfate (Jimenez et al., 2009). As a comparison, the OOA profile is poorly correlated with semi-volatile OOA (SV-OOA; R$^2$ of 0.34 between the OOA and the reference SV-OOA profile) (Ng et al., 2011b). However, the diurnal pattern of OOA in Dublin showed a clear pattern that was strongly influenced by local sources and was most likely from fresh SOA instead of well-aged SOA. Moreover, the morning peak of OOA came about 1 hour later than that for HOA, probably indicating fast SOA formation processes by atmospheric oxidation of SOA precursor gases and/or condensation of semi-volatile VOCs emitted by the nearby traffic. Besides, the contribution from cooking sources to OOA was also important (Liu et al., 2018) as evidenced by the concurrent peaks of OOA with COA from their diurnal patterns. The diurnal trend of OOA suggests the OOA had a background concentration of 0.5 µg m$^{-3}$ which was higher than HOA (0.1 µg m$^{-3}$) and COA (0.1 µg m$^{-3}$), indicating part of OOA was also associated with regional transport. It is estimated that approximately 38% of the total OOA was regionally transported, i.e., the value of background OOA concentration of 0.5 µg m$^{-3}$ if compared to the total OOA concentration of 1.3 µg m$^{-3}$.

### 3.4.2    OA factors at the residential site during non-heating

Free PMF suggested a two-factor solution with one mixed primary OA factor and one OOA (Fig. S7) as a further increase in the number of factors led to the splitting of factors. The diurnal pattern of the primary factor showed a slight increase in the morning but with a larger increase in the evening and night, suggesting potential mixing between HOA and the heating-related factor. Note that the temperature was below 15 °C in the evening during this period (Fig. S3) and thus sporadic domestic solid fuel burning activities were likely to occur. Our previous study has shown that peat burning occurred in cold summer nights in the west coast city of Galway in Ireland (Lin et al., 2019a). Moreover, the elevated levels of m/z 60 in the evening (at 22:00) suggested emissions from biomass burning (Fig. S7). The profile of the unconstrained primary OA factor is better correlated with the reference profile of peat burning OA (R$^2$= 0.58) than that of wood burning OA (R$^2$=0.20) from our previous study (Lin et al., 2017) and the averaged profile of biomass burning OA (BBOA; R$^2$=0.44) factor from the Ng et al. (2011a)." Note that COA was not considered to be a potential OA factor at this location since the sampling site was representative of the residential area with few restaurants around. Moreover, no clear increase in the concentration of POA during lunchtime was observed during this non-heating period and the heating period as discussed later. Therefore, only the reference profile of HOA and peat burning OA factors were constrained during non-heating at the residential site.

The mass spectra and diurnal patterns, as well as the relative contribution of the HOA, peat, and OOA at the residential site during non-heating are shown in Fig. 6. While the profile of HOA is similar between the residential site and the kerbside, its concentration levels at the residential site were significantly lower than at the kerbside. Specifically, in the morning rush hours, the HOA peak concentration was 0.1 μg m$^{-3}$ at the residential site while it was 1.5 μg m$^{-3}$ at the kerbside, with a 15 times difference between the two sites. On average, HOA accounted for 11.8% of OA at the residential site. During the morning rush hours, its fraction increased to 16.0% which was still a minor factor when compared to OOA (58-68% of OA). The low contribution of HOA is consistent with the low mixing ratio of NO$_x$ at the residential site (median: 2.0 ppb) which was 20 times lower than that at the kerbside (Table 1).

The peat profile featured peaks at *m/z* of 27, 29, 41, 43, 55, and 57, which was similar to HOA. However, the differences in *f60* between the peat factor and HOA suggested different sources. *f60* in the peat profile was 0.014 which was higher than that for HOA (<0.003), confirming its biomass nature (Cubison et al., 2011). The diurnal pattern of peat showed increased concentrations at 20:00-22:00, corresponding to their emission time. On average, peat accounted for 23.6% of OA and its fraction increased to 28.1% in the evening. The OOA mass spectral at the residential site featured high contribution at m/z 44 (i.e., *f44*) which was similar to the OOA (R$^2$=0.95 between the two OOA total mass spectral profile) at the kerbside. Compared to that at the kerbside, the diurnal cycle of OOA at the residential site showed only slightly elevated concentration in the evening (Fig. 6b). Therefore, most of the OOA at the residential site was likely due to regional transport.

### 3.4.3 OA factors at the kerbside during heating

Free PMF solutions found the concentration of the heating-related OA factor increased substantially in the evening (Fig. S8). The increase in OA concentration during the evening was likely due to the emissions from the use of oil, peat, coal, and wood in the domestic sector according to Central Statistics Office (CSO, 2016) and Sustainable Energy Authority of Ireland (SEAI, 2018). Previous studies have identified these heating-related aerosols in urban areas in Ireland (Kourtchev et al., 2011; Dall'Osto et al., 2013; Lin et al., 2018). However, due to the temporal covariation of the heating-related OA factors, free PMF led to highly mixed factors which were insufficient to evaluate their respective contributions. ME-2 was, thus, used to constrain the reference profiles of peat, coal, and wood from Lin et al. (2017), along with HOA and COA (Crippa et al., 2013).

Figure 7 shows the mass spectral profiles, diurnal patterns, and relative contributions of the resolved OA factors at the kerbside during heating. Consistent with that during non-heating, the HOA factor also showed rush hour peaks in the morning (8:00) and afternoon (17:00) with similar concentrations (1.8 μg m$^{-3}$) during heating to that during non-heating. However, a third HOA peak (2.2 μg m$^{-3}$) was also seen at ~22:00 (Fig. 7b), likely from the emission of oil burning (CSO, 2016; Lin et al., 2017; Lin et al., 2018). Over the entire heating period, HOA accounted for 15.7% (1.2 μg m$^{-3}$) of OA (Fig. 7c). The relatively small fraction of HOA indicates traffic was not the dominant OA source for the heating period despite measurements being adjacent to the busy road in downtown Dublin. Note that HOA during heating could only be taken as an upper limit of the primary OA emissions from traffic because of the additional contribution from the oil burning (discussed in Sect. 3.5). COA showed a similar diurnal pattern to that during non-heating but with slightly higher concentrations during the heating period,

which could be associated with colder temperature and shallower boundary layer. On average, COA accounted for 17.9% (1.2 µg m$^{-3}$) of OA (Fig. 7c).

Peat is an accumulation of partially decayed plant material which is an important domestic fuel source in Ireland (Tuohy et al., 2009). The incomplete decay of vegetation resulted in an increase of *f60* when burned (Lin et al., 2017). However, *f60* in peat profile (0.014) was lower than wood (0.053) probably because wood contained a higher fraction of *m/z* 60-related material e.g., levoglucosan (Fig. 7a). Over the entire heating period, peat burning accounted for 17.7% (1.2 µg m$^{-3}$) of OA (Fig. 7c). However, during the pollution spikes as seen on 28 and 31 October (Fig. 7d), peat burning increased its fraction to 26.5% (9.2 µg m$^{-3}$), suggesting peat burning was an important OA emission source. Similarly, wood burning increased its contribution to 9.8% (3.3 µg m$^{-3}$) during pollution spikes from an average of 7.3% (0.5 µg m$^{-3}$). The important role of peat and wood burning in driving the pollution events is consistent with our previous study in suburban Dublin (Lin et al., 2018).

The profile of coal burning OA featured very low *f60* (<0.003), consistent with its non-biomass signature as coal is formed from the complete vegetation decay. On average, the coal factor accounted for 12.6% (0.8 µg m$^{-3}$) of OA. Though the fraction of coal burning decreased during pollution events (11.9%; Fig. 7d), its absolute concentration increased (4.3 µg m$^{-3}$). Note that chloride also showed a significant increase during the pollution events (Fig. 2c) which was associated with coal burning as our previous coal-combustion experiment showed that chloride emission comprised a high fraction (2-52.8%) of the submicron aerosol from coal burning emissions (Lin et al., 2017).

The OOA profile had an *f44* of 0.24 during heating which was similar to that (0.22) during non-heating, indicating similar oxidation levels between the two periods. However, compared to the daytime OOA peak concentrations (1.8 µg m$^{-3}$), higher OOA concentrations (3.9 µg m$^{-3}$) were observed in the evening, indicating a more important contribution from the condensation of SVOCs emitted from heating sources and/or their dark aging processes (Tiitta et al., 2016). On average, OOA accounted for a large fraction (29% or 1.9 µg m$^{-3}$) of OA. During pollution events, OOA still accounted for a significant fraction of OA (22% or 7.8 µg m$^{-3}$), demonstrating its importance of secondary OA processing.

### 3.4.4 OA factors at the residential site during heating

Heating-related OA factors were identified since they all showed elevated concentrations in the evening as indicated by the free PMF solutions (Fig. S9). However, similar to the case at the kerbside, the OA factors were mixed because of the co-emissions from all domestic heating activities. To better evaluate the contribution of potential sources, the reference profiles of HOA (Crippa et al., 2013), peat, wood, and coal were constrained (Lin et al., 2017) using ME-2. Note that COA was not constrained since no lunch meal peaks were identified during this period as discussed above.

In the morning rush hours, HOA showed a peak concentration of 0.4 µg m$^{-3}$ due to traffic emissions, which was, again, largely reduced when compared to the HOA morning peaks (1.8 µg m$^{-3}$) at the kerbside. However, higher concentrations of HOA (2.0 µg m$^{-3}$) were observed in the evening due to the emissions of oil burning. Therefore, the majority (estimated at over 90%) of HOA at the residential site was due to oil burning instead of traffic emissions. On average, HOA accounted for 11.8% of the total OA over the entire heating period and its fraction increased slightly to 12.9% during pollution events.

Similar to the kerbside, solid fuel burning, especially peat burning was a very important OA factor, contributing up to 45.5% (34.9% on average) of the OA during the pollution events (Fig. 8c and 8d). Coal (17.8-19.2%) and wood (7.1-7.5%) burning were also contributing substantially over the entire period as well as during pollution events. The importance of solid fuel burning at the residential site echoed our previous studies at the same sites (Lin et al., 2018; Lin et al., 2019b). OOA, on average, accounted for 27.9% (1.4 μg m$^{-3}$) of OA and the higher OOA concentration during the evening, concurrent to an increase in primary factor concentrations, again suggested its major source was from the condensation of SVOCs emitted from solid fuel burning and/or their dark aging processes (Tiitta et al., 2016).

## 3.5     Implications for emission control

The HOA/BC ratio was ~0.17 as retrieved from the slope of the linear regression between HOA and BC with a coefficient of determination ($R^2$) being 0.61 for the non-heating period at the kerbside (Fig. S10). This points to, on average, 6 times higher traffic contribution to BC as compared to HOA.. Note that HOA represented only the primary fraction of the organic aerosol emitted from the traffic. The secondary formation from the oxidation of precursor gaseous emissions from traffic was also important, as indicated by the elevated OOA concentration in the morning rush hours at the kerbside (Fig. 5b). This is consistent with the smog chamber studies by Gordon et al. (2014), which showed substantial amount of secondary OA formation from the precursor gases emitted by diesel vehicles with no aftertreatment techniques. In this study, the morning OOA peaks at the kerbside were mostly coinciding with the increase in HOA, pointing to a fast formation of secondary organic aerosol (i.e., from minutes to hours). However, the latter requires further investigations (e.g., real-time measurement of its precursor gases, oxidants etc.) to better understand the formation pathways in a real urban environment.

Regarding the HOA to BC ratio for the vehicular emissions, only the period of non-heating at the kerbside was considered. This is because HOA and BC were exclusively from traffic during this period. Note that the contribution from cooking to the ambient BC was considered negligible as there was a lack of BC peak during the lunch time at the kerbside (Fig. 4a). The negligible BC emissions from cooking were consistent with previous studies e.g., by Lee et al. (2015) and He et al. (2019). In contrast, during the heating period, HOA and BC had other sources (i.e., heating) in addition to traffic. For example, while both HOA and BC showed typical two rush hour peaks, an additional peak in the evening was observed during the heating period. Such evening peaks of HOA during the heating period could not be associated with vehicular emissions since, during the non-heating period, HOA showed only morning and afternoon rush-hour peaks. Also, the traffic pattern (i.e., with no increase in traffic in the evening) was not expected to change from the non-heating to the heating period. Instead, as discussed in Sect. 3.4, the evening peaks of HOA were associated with the use of oil for heating. Since the average diurnal profile of HOA, during the heating period, showed two rush hour peaks as during non-heating period and one additional evening heating peak, 67% of HOA was apportioned to be from vehicular emissions and 33% from the oil heating. As a comparison, at the residential site during the heating period, HOA was nearly exclusively (95%) from heating because it showed substantially higher evening peak than the morning rush hour peak.

The HOA/BC ratio in Dublin was lower than the HOA/BC ratios reported for the gasoline vehicles-dominated environment (0.9-1.7) but was within the range for the diesel environment (0.03-0.61) (DeWitt et al., 2015), indicating most of the traffic emissions were from diesel vehicles. The large increment of HOA and BC at the kerbside in the Dublin city centre holds important implication for its adverse health impacts since the International Agency for Research on Cancer associated the exposure to the diesel engine exhaust with an increased rate of lung cancer. In Ireland, diesel fuel accounted for 73% of the

on-road transport energy in 2017 (Fig. S11a). Figure S11a also shows an increasing trend of diesel fuel usage, indicating worse air quality in the predictable future at the kerbside if diesel vehicular emissions were not controlled.

    This study also shows that vehicular emissions appear to impact the air quality adjacent to the roads while, in contrast, solid fuel burning has a large geographic impact, affecting overall air quality at the kerbside and residential sites examined in this investigation. Such large geographic impact suggests significant climate effects from residential emissions (Butt et al.,

2016) which contain a higher fraction of OA than that from traffic. However, surprisingly, the census data shows only a few households (<5%) consuming these solid fuels with the majority households (~95%) using natural gas and electricity as the primary heating sources in 2016 (Fig. 11b). Therefore, if the emissions from this small fraction of households were controlled, good overall air quality and lower climate forcing can be expected.

## 4     Conclusion

The chemical composition and sources of submicron aerosol ($PM_1$) were simultaneously investigated at a kerbside location in downtown Dublin and at a residential site in south Dublin (~5 km apart) using an ACSM and AE33/16 at both sites during both non-heating (i.e., early September) and heating periods (i.e., late October) of 2018. Traffic emissions were found to have a minor impact on air quality at the residential site due to the distance from nearby roadways and other affecting parameters such as wind speed and wind direction. In contrast, the kerbside was found to be highly affected by diesel vehicular emissions.

BC was the most dominant component (38-55%) of $PM_1$ at the kerbside location while OA was the most important species (46-64% of $PM_1$) at the residential site. During non-heating period, an increment ratio of up to 25 was found for the BC at the kerbside when compared to the level of BC at the residential site primarily due to vehicular emissions. During the heating period, the BC increment ratio was lower (~10) due to the additional sources of solid fuel burning which contributed to the BC concentrations at both sites. Moreover, solid fuel burning was shown to increase $PM_1$ concentrations substantially with episodic

concentrations of >100 $\mu g\ m^{-3}$ being recorded at both sites simultaneously. Source apportionment of OA using ME-2 showed that only 16-28% (upper limit due to the additional heating source of HOA) of OA could be directly associated with vehicular emissions (i.e., HOA) at the kerbside, with the larger contribution of OA being attributed to cooking, solid fuel burning, and OOA. HOA contributed to 11-12% of OA at the residential site with a large contribution from oil burning instead of traffic because HOA did not show elevated rush hour peaks as found at the kerbside. During the heating period, solid fuel burning

contributed to over 50% of OA at the residential site, while oxygenated OA accounted for almost 65% of OA during the non-heating period. This study highlights the significant increment of BC due to the traffic emissions at the kerbside and the large

geographic impact of OA from residential heating at both the kerbside and residential sites. Therefore, traffic and residential heating might have different health and climate implications as suggested by the temporal and spatial variability of sources within Dublin city centre.

**5    Data Availability**

All data needed to evaluate the conclusions in the paper are present in the paper and/or the Supplementary Materials. Also, all data used in the study are available from the corresponding author upon request.

**6    Author Contribution**

JO, DC, SJ, and CO'D conceived and designed the experiments; CL, JO, DC, and PB performed the experiments; CL, JO, 490 WX, EH, JW, JG, and CO'D analyzed the data; CL prepared the manuscript with input from all co-authors.

**7    Competing interests**

The authors declare that they have no conflict of interest.

**8    Acknowledgments**

This work was supported by EPA-Ireland (AEROSOURCE, 2016-CCRP-MS-31), Department of Communications, Climate 495 Action and Environment (DCCAE), the National Natural Science Foundation of China (NSFC) under grant no. 41925015, 91644219           and           41877408,           the Chinese Academy of Sciences (no. ZDBS-LY-DQC001,  XDB40030202), the Cross Innovative Team fund from the State Key Laboratory of Loess and Quaternary Geolog y (SKLLQG) (no. SKLLQGTD1801), and the National Key Research and Development Program of China (no. 2017YFC02 12701). The authors would also like to acknowledge the contribution of the COST Action CA16109 (COLOSSAL) and MaREI, 500 the SFI Research Centre for Energy, Climate and Marine.

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

Table 1: Mean, standard deviation (SD), first quartile (Q1), median, third quartile (Q3), and maximum (Max) concentration of the hourly averaged organic aerosol (OA), sulfate ($SO_4$), nitrate ($NO_3$), ammonium ($NH_4$), chloride (Chl), and black carbon (BC) at the kerbside and residential site during the Non-heating period (10/09 – 17/09) and Heating period (27/10 – 03/11 2018). Aerosol species are in µg m$^{-3}$ and NOx is in ppb.

| Non-Heating | Kerbside (10/09 - 17/09 2018) | | | | | | Residential (10/09 – 17/09 2018) | | | | | |
|---|---|---|---|---|---|---|---|---|---|---|---|---|
| | mean | SD | Q1 | median | Q3 | Max | mean | SD | Q1 | median | Q3 | Max |
| OA | 3.3 | 2.1 | 1.8 | 3.0 | 4.3 | 8.7 | 1.0 | 0.7 | 0.5 | 0.9 | 1.4 | 3.3 |
| $SO_4$ | 0.5 | 0.4 | 0.2 | 0.4 | 0.7 | 1.6 | 0.4 | 0.2 | 0.2 | 0.3 | 0.5 | 2.6 |
| $NO_3$ | 0.3 | 0.2 | 0.2 | 0.2 | 0.3 | 0.7 | 0.1 | 0.1 | 0.1 | 0.1 | 0.1 | 0.4 |
| $NH_4$ | 0.4 | 0.5 | 0.1 | 0.3 | 0.7 | 1.4 | 0.3 | 0.4 | <0.1 | 0.2 | 0.5 | 1.4 |
| Chl | 0.1 | 0.3 | <0.1 | <0.1 | 0.2 | 1.0 | <0.1 | <0.1 | <0.1 | <0.1 | <0.1 | 0.1 |
| BC | 5.6 | 4.6 | 2.6 | 4.6 | 7.3 | 18.2 | 0.4 | 0.4 | 0.2 | 0.3 | 0.4 | 3.0 |
| $PM_1$[a] | 10.3 | 5.1 | 6.2 | 9.7 | 13.5 | 21.2 | 2.3 | 1.3 | 1.3 | 2.0 | 2.8 | 8.5 |
| $NO_x$ | 54.1 | 42.7 | 17.4 | 45.1 | 79.9 | 200 | 2.6 | 2.3 | 0.8 | 2.0 | 4.0 | 12.2 |
| Heating | Kerbside (27/10 – 03/11 2018) | | | | | | Residential (27/10 – 03/11 2018) | | | | | |
| OA | 6.5 | 7.9 | 1.4 | 3.7 | 8.3 | 45.5 | 8.1 | 15.5 | 0.9 | 2.3 | 7.0 | 90.0 |
| $SO_4$ | 0.9 | 1.4 | 0.2 | 0.5 | 1.1 | 8.6 | 0.9 | 1.3 | 0.2 | 0.4 | 1.0 | 8.4 |
| $NO_3$ | 1.6 | 1.9 | 0.2 | 0.7 | 2.6 | 8.3 | 1.1 | 1.4 | 0.1 | 0.4 | 1.5 | 7.8 |
| $NH_4$ | 1.3 | 2.5 | 0.1 | 0.7 | 1.5 | 18.1 | 1.0 | 1.7 | 0.2 | 0.5 | 1.3 | 11.6 |
| Chl | 1.0 | 3.9 | <0.1 | 0.1 | 0.4 | 28.5 | 0.6 | 2.1 | <0.1 | 0.1 | 0.2 | 18.3 |
| BC[b] | 7.1 | 8.3 | 1.4 | 3.8 | 9.6 | 37.5 | 1.2 | 2.4 | 0.2 | 0.4 | 1.0 | 17.0 |
| $PM_1$ | 18.4 | 18.5 | 5.2 | 12.3 | 24.9 | 134.0 | 12.7 | 23.3 | 2.6 | 5.7 | 14.8 | 126.0 |
| $NO_x$ | 89.7 | 85.5 | 28.5 | 60.3 | 123.5 | 410 | 16.5 | 22.6 | 3.6 | 8.2 | 18.0 | 139.0 |

[a]$PM_1$ sum of OA, $SO_4$, $NO_3$, $NH_4$, Chl and BC

[b]BC at the residential site was not available during non-heating period and the week after (from 24 September to 30 September) was shown as a reference.

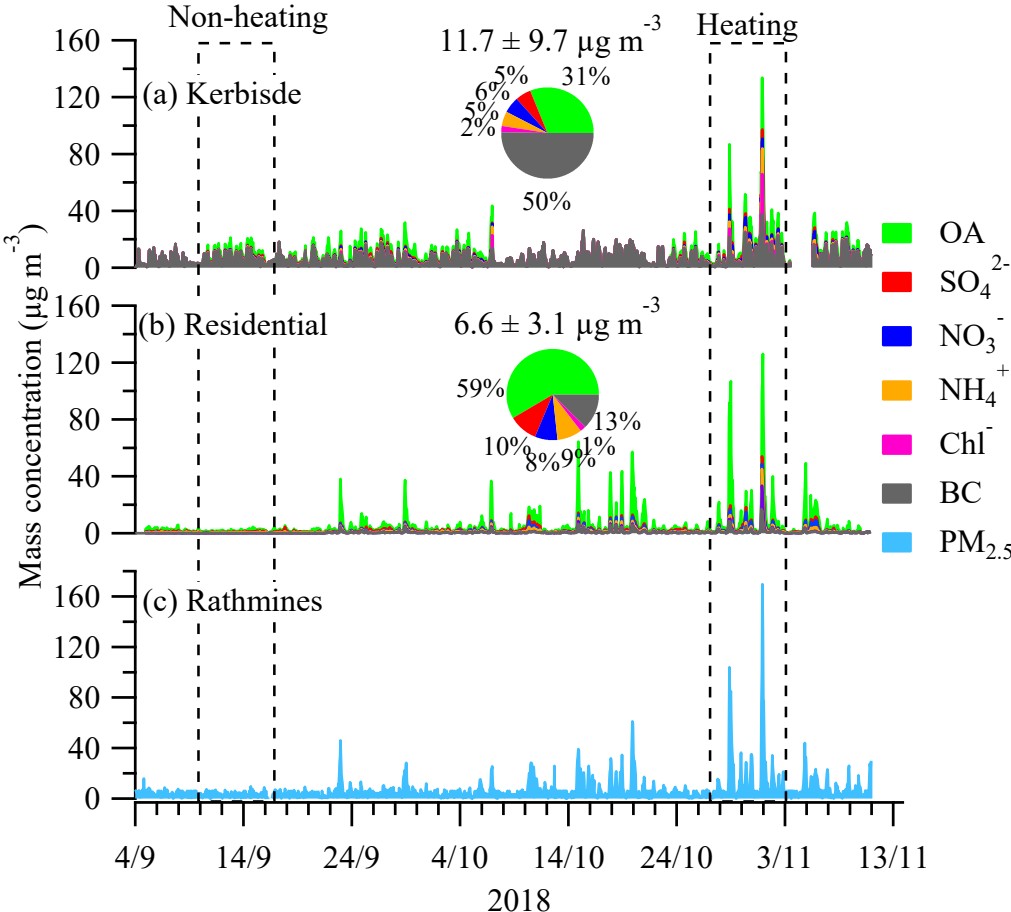

**Figure 1.** Time series of hourly averaged submicron organic aerosol (OA), sulfate ($SO_4^{2-}$), nitrate ($NO_3^-$), ammonium ($NH_4^+$), chloride ($Chl^-$), and black carbon (BC) at the kerbside (a), residential site (b), and EPA Rathmines station (c) from 4 September to 9 November 2018. Inset pie charts are the chemical composition of $PM_1$ averaged over the entire period while the numbers ($\pm$one SD) above are the average $PM_1$ concentration. The non-heating period, from 10 September to 17 September, and heating period, from 27 October to 4 November 2018 are marked for further analysis. The ACSM data gaps at the kerbside (e.g., from 7/10 to 23/10) were due to ACSM malfunction.

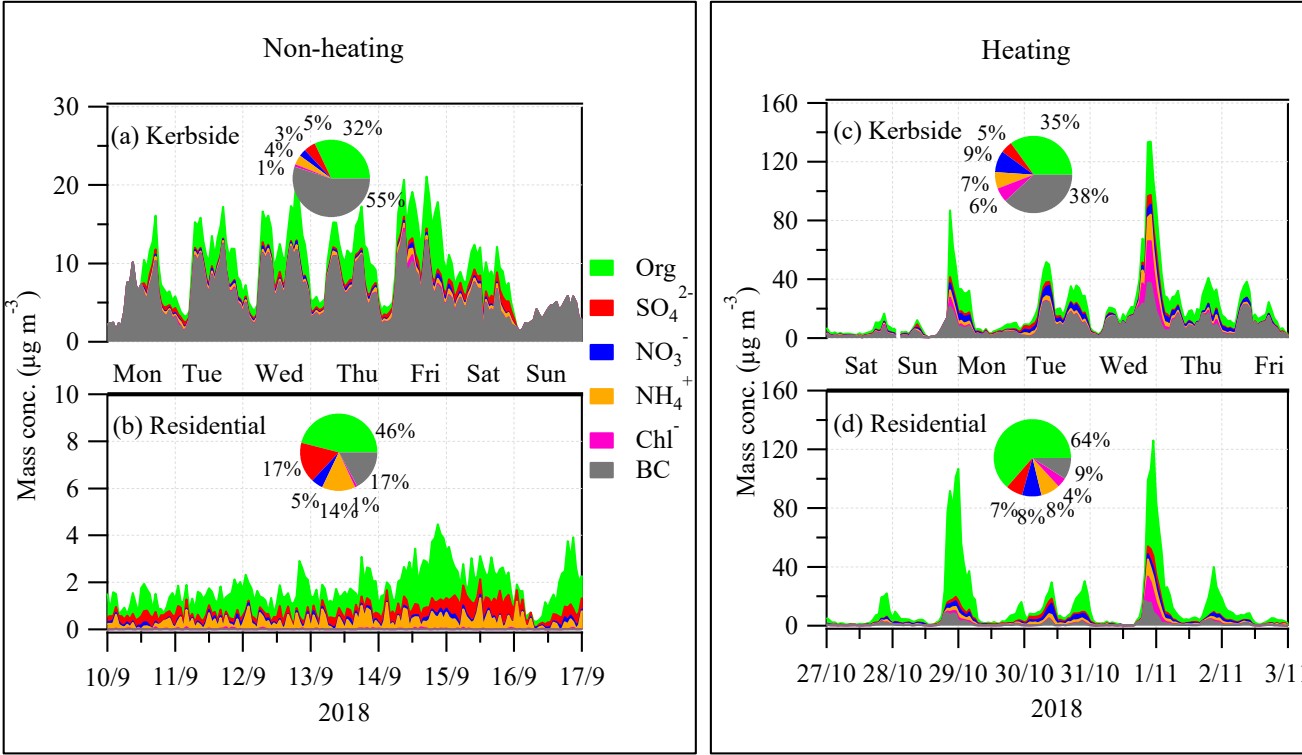

**Figure 2. Time series of hourly averaged submicron organic aerosol (OA), sulfate (SO₄), nitrate (NO₃), ammonium (NH₄), chloride (Chl), and black carbon (BC) during non-heating (left panel) and heating (right panel) at the kerbside (a, c) and residential (b, d). Inset pie charts are the relative fraction of the measured PM₁ components. Also shown is the day of the week, including Monday (Mon), Tuesday (Tue), Wednesday (Wed), Thursday (Thu), Friday (Fri), Saturday (Sat), and Sunday (Sun). At the kerbside, the**

685 **ACSM data gaps (e.g., on 16/9) were due to ACSM malfunction. At the residential site, BC was not available during the non-heating period. As a reference, the average concentration from the week later was included for the calculation of relative contribution to PM₁.**

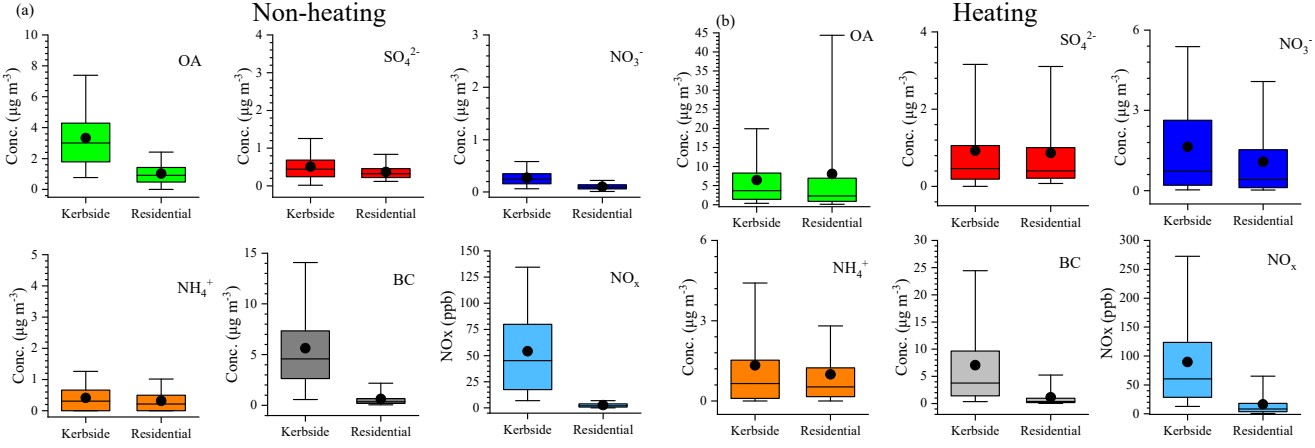

**Figure 3. Box plots of the submicron organic aerosol (OA), sulfate (SO₄²⁻), nitrate (NO₃⁻), ammonium (NH₄⁺), black carbon (BC), and NOₓ during non-heating (a) and heating (b) at the kerbside and residential site. The median, the 25th and 75th percentiles are represented by the middle, lower and upper vertical bars, respectively. The 5th and the 95th percentiles are the bottom and top whiskers, respectively.**

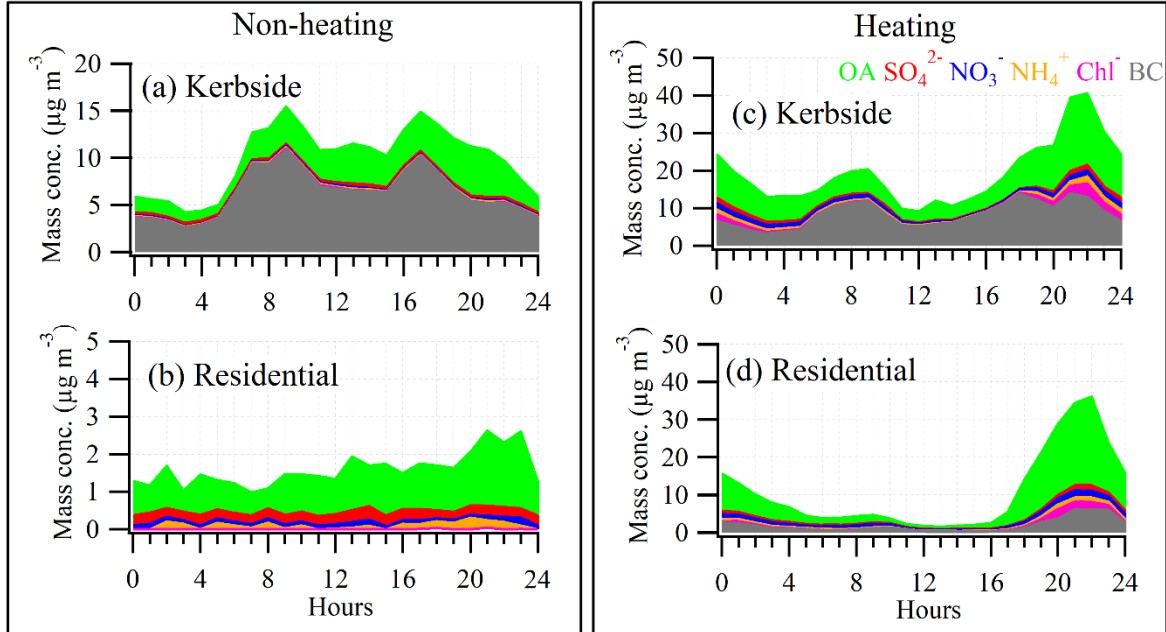

**Figure 4. The diurnal cycle of submicron organic aerosol (OA), sulfate (SO₄²⁻), nitrate (NO₃⁻), ammonium (NH₄⁺), chloride (Chl⁻), and black carbon (BC) during non-heating (left panel) and heating (right panel) at the kerbside (a, c) and residential site (b, d).**

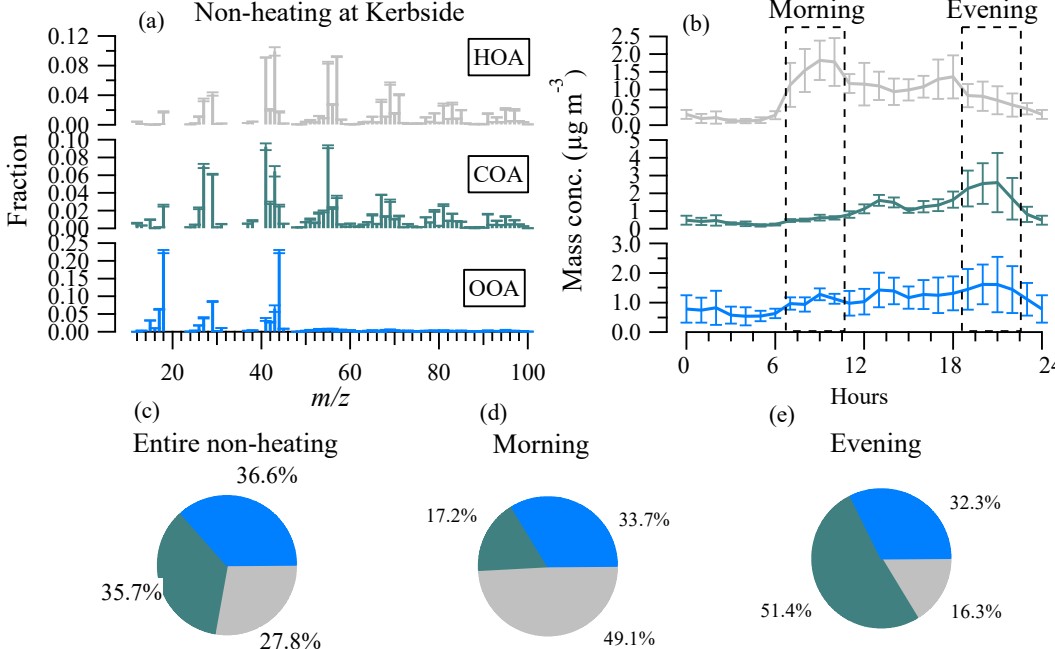

Figure 5. (a) Mass spectral profiles of hydrocarbon-like OA (HOA), cooking OA (COA) and oxygenated OA (OOA) during non-heating at the kerbside; (b) Diurnal cycle of the OA factors; and (c) relative contribution of OA factors to the total OA over the entire non-heating period; (d) over the morning rush hours (7:00-10:00, local time); and (e) over the evening hours from 19:00-22:00. Error bars in (a) and (b) represent one standard deviation.

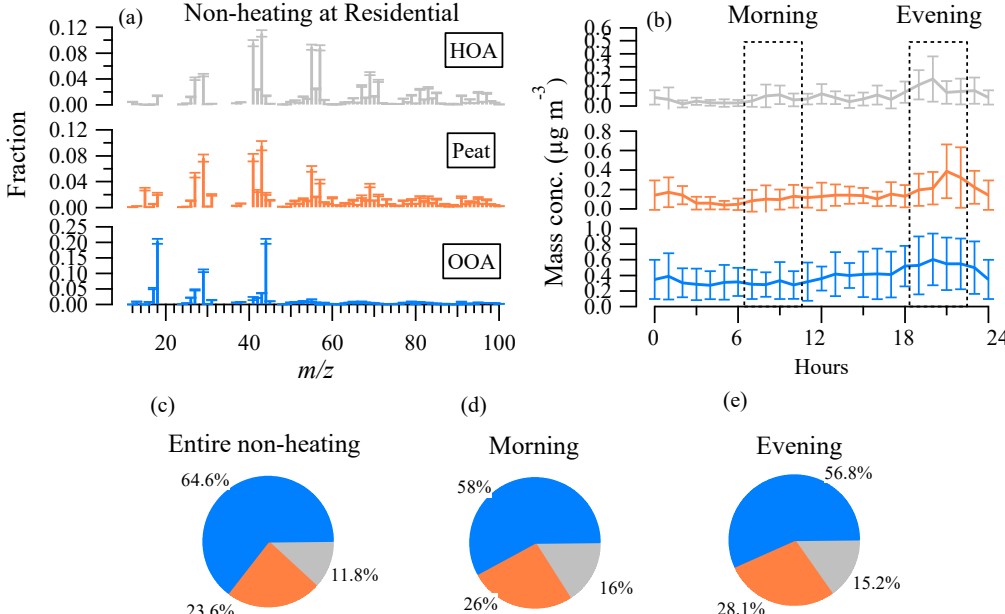

Figure 6. (a) Mass spectral profiles of hydrocarbon-like OA (HOA), peat and oxygenated OA (OOA) during non-heating at the residential site; (b) Diurnal cycle of the OA factors; and (c) relative contribution of OA factors to the total OA over the entire non-heating period; (d) over the morning rush hours (7:00-10:00); and (e) over the evening hours from 19:00-22:00. Error bars in (a) and (b) represent one standard deviation.

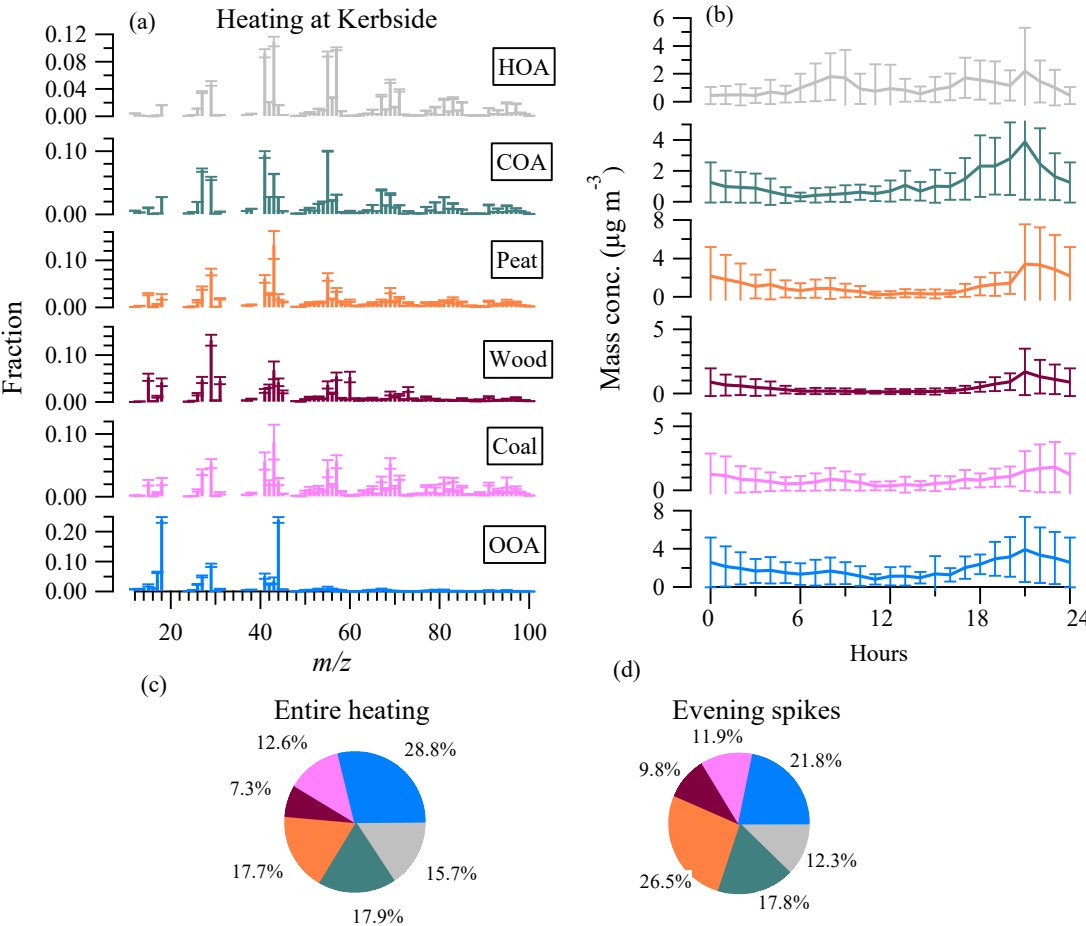

**Figure 7. (a) Mass spectral profiles of hydrocarbon-like OA (HOA), cooking OA (COA), peat, wood, coal and oxygenated OA (OOA) factors during heating at the kerbside; (b) Diurnal cycle of the OA factors; and (c) relative contribution of OA factors to the total OA over the entire heating period; and (d) over evening spikes on 28 and 31 October. Error bars in (a) and (b) represent one standard deviation.**

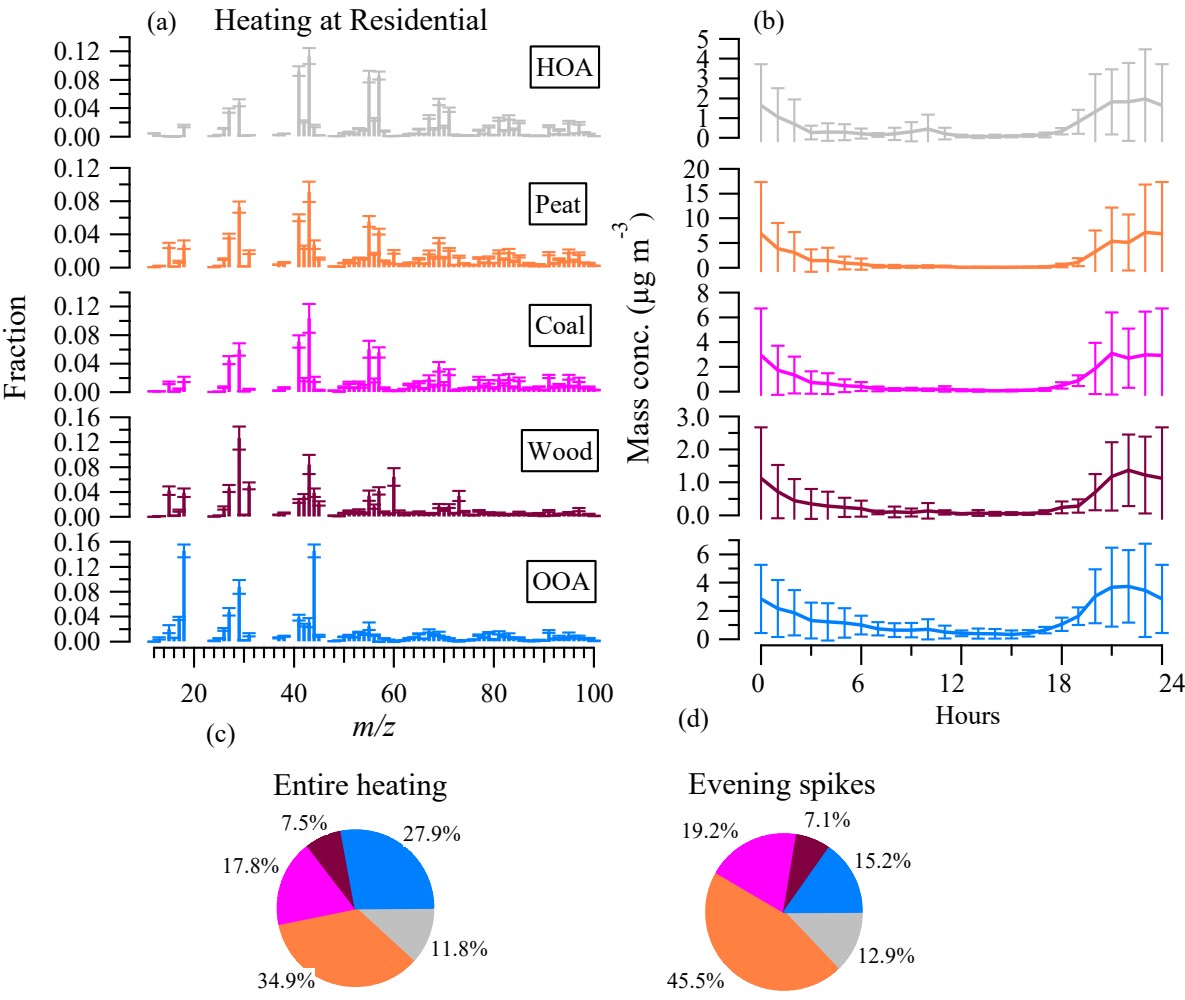

**Figure 8. (a) Mass spectral profiles of hydrocarbon-like OA (HOA), peat, wood, coal and oxygenated OA (OOA) factors during heating at the residential site; (b) Diurnal cycle of the OA factors; and (c) relative contribution of OA factors to the total OA over the entire heating period; and (d) over evening spikes on 28 and 31 October. Error bars in (a) and (b) represent one standard deviation.**
