# Peer review of "The impact of traffic on air quality in Ireland: insights from the simultaneous kerbside and sub-urban monitoring of submicron aerosols"

_Atmospheric Chemistry and Physics, 2019_

## Referee Comment (RC1) · Anonymous Referee #1 · 2 Mar 2020

General comments:

In this study, Lin et al. evaluated the chemical composition and organic aerosol (OA) sources of PM1 monitored simultaneously at a kerbside site and a residential site in Dublin during both non-heating and heating periods. The authors found that vehicle emissions were associated with the significant kerbside increment of black carbon during the non-heating period, but they had a minor impact on air quality at the residential site. Significant contributions of solid fuel burning to OA were observed at both sites during the heating period.

The findings in this study provided valuable information for aerosol scientists to better

understand the temporal and spatial variations of the concentration, chemical composition, and OA origins of PM1 in Ireland, which could help the formulation of air quality policies and PM1 mitigation strategies in Ireland. Moreover, the paper is well written, and the results are visualized in an appropriate way. Therefore, I recommend it for publication after minor revisions.

Specific comments:

1. Line 14, "...sources of submicron aerosols (PM1)..." Actually, the authors performed the source analysis of OA instead of PM1 in this study. Please revise.

2. I think it is more straightforward to use "the non-heating period" and "the heating period" instead of "P1" and "P2" in the manuscript, especially in the figures and tables.

3. Line 149 and 170, the spikes of BC and OA were defined as those with measured concentrations higher than 15 $\mu$g/m3 and 5 $\mu$g/m3, respectively. What are the criteria to set these two threshold values?

4. Line 186. As mentioned by the authors, the dominant wind direction during the non-heating period was southwest while the city center was located to the north of the residential site. Was there any connecting flow between these two sites? If so, how did the BC concentration vary at the two sites under the connecting flows?

5. Line 194, Fig. 1c and 1d, another peak concentration was shown in the evening on 29 October, 2018. The authors suggested that these should be due to the residential heating activities. Their impacts on PM1 at residential site are comparable on both dates. However, the burning activity on 31 October showed a greater impact on PM1 than that on the other date at kerbside site. Do the authors have any idea what these burning activities are? Do these kinds of burning activities usually happen during the heating period in Ireland or just special for this sampling period? Are they open-field biomass burning? The reason for me to ask about this is obviously these two burning activities have pulled up the mean values of PM1, OA, BC, NOx etc. at the two

sites during the heating period. More information about these burning activities will be helpful for the data interpretation and discussion in line 243-250.

6. Line 234, in addition to the low temperatures in the evening, the PBL during the evening was usually lower than that during the day, and thus PBL could also partly contribute to the higher concentration of COA.

7. Line 241, could the authors estimate the fraction of OOA that were regionally transported to the kerbside site?

8. Line 296, it is not clear what are the typical values of HOA/BC for the gasoline vehicles-dominated environment. Please add relevant values and references.

9. Was the second ACSM also equipped with a quadrupole mass spectrometer? How were the Q-ACSMs calibrated?

10. Table 1 & 2 can be combined into one. Please add PM1 data in the table. Moreover, numbers in the Table should have the same significant figures as those in the main text of the manuscript and figures.

Several typos:

11. Fig. 4d, the OA concentration should be 8.1 instead of 8.7 $\mu$g/m3.

12. Line 43, "micron-environment" should be "micro-environment".

13. Line 133, should be Crippa et al. (2013).

---

## Referee Comment (RC2) · Anonymous Referee #2 · 7 Apr 2020

Lin et al. (2020) in this study measured the chemical composition of submicron aerosol (PM1) in two locations in Dublin (kerbside and residential sites) from 4 September to 9 November 2018. Measurements were performed by using an Aerodyne Aerosol Chemical Speciation Monitor (ACSM) and an Aethalometer (AE-16 and AE-33). PMF analysis was also implemented on the data.

In general, the manuscript is well written though the results presented in the main manuscript do not correspond to the entire period of 4 September to 9 November 2018, but to only two weeks of measurements. The authors chose two periods of the available data, with the first one corresponding to a rather clean period (P1) with

low mass concentration of PM1 and another one rather polluted (P2) by local sources (heating purposes).

The study presented is very rich in data and analysis. However, these information are not very clear in the main manuscript or not even included (they are left on the supplementary) making the manuscript weak and less readable, especially in the results section. Additionally, the reading might be challenging as OA sources and changes in concentrations are vaguely discussed and not really explained. Moreover, figures need reshaping as labels are not well positioned or data are not clear due to high values of the axis.

Specific comments:

Line 26: "solid fuel burning..." it should be explained for which period is referred to. Solid fuel burning was not present in both periods.

Line 77: the word "monitor" is missing.

Line 107: The 1 hour interval that was set for ACSM in the residential site is really high. This might lead to the lack of data originating from local sources and may affect the corresponding diurnal profiles of OA sources. Authors chose this in order to reduce uncertainty though in Fig. S1b the measured PM1 mass concentration is quite high reaching values greater than the Kerbside site.

Line 110: How much the collection efficiency was on average?

Line 111: The Rathimnes site is mentioned. It was not described before in the manuscript and is referred also afterwards without any information. Please provide some info.

Line 118: CO measurements are available for kerbside but they are not presented anywhere in the main manuscript.

Line 137: Figure S1 deserves to be in the main manuscript. The overall (2 months

period) chemical composition of submicron aerosol (PM1) should also be included for both sites and discussed.

Line 151: Please specify Fig.1 and Fig.2 with Fig.1a, Fig.2a.

Line 168: Fig.3a is mentioned before Fig. 1b, c,d and Fig. 2b, c, d. Please fix this.

Line 173: "(Fig. 1)" Please define to which letter of Fig. 1 the text is referring to.

Line 195: "197.3 $\mu$g m-3" This value is not consistent for Fig.1 c and Fig. S1a. Please provide the correct figure/ value.

Line 207: Nitrate can be also present in higher values due to organonitrates (ON). Have the authors checked the fraction of ON to nitrate?

Line 215: "(Fig. S1)" should change to Fig. S1c

Line 226: Fig. 5a is referred before Fig. 4b, c, d.

Line 234: Higher COA mass concentrations could be related to the decrease of boundary layer.

Line 235: Fig. S7 does not correspond to the OOA profile spectrum. Fig. S6a, b deserves to be in the main manuscript instead of Fig. 5a.

Line 235: The authors state that OOA resembles of the less volatile OOA (LV-OOA). What criteria lead to this statement?

Line 243: The authors have constrained with ME-2 tool, 4 out of 5 factors with a relative strict $\alpha$-value. This leads to two factors (wood and coal) contributing fairly low to OA (4% and 8%, respectively), making the results less robust. How confident are the authors about this constraint? It seems that solution was forced to an erroneous solution, comparing the data to the unconstrained ones, where biomass burning factor contributed for approx..20% of OA.

Line 244: Fig. 4c and Fig. 5a are referred before Fig. 4b and Fig. 5b.

Line 245: There is no such high peak in HOA around 23:00, according to Fig. S11b. It seems that on Fig. 5 the 3rd quartile of the data has been used instead of the average values. This is consistent to the average values given in the manuscript. Again, Fig. S11 deserves to be in the main text instead of Fig. 5c.

Line 253: peat, coal and wood factors contribute one third to the total OA during P2 in kerbside. However, only a line is describing their existence.

Line 254-255: Could this higher OOA levels be transported by another area and not due to SVOCs?

Section 3.3.2: Lacks of discussion.

Line 258: A peat and a HOA factor were constrained with ME-2. Why the authors chose a peat factor instead of a BBOA one from literature? Does this change the results significantly? Also, a COA factor is not present. Have the authors tried to constrain it? In this case how much was its contribution to the OA? Again Fig. S14a,b should be included in the main text.

Line 265: Again 4 out of 5 factors have been constrained with ME-2 tool. The unconstrained solution of this period (P2) for the residential site is not given in the supplementary so no direct comparisons can be made. How confident are the authors about this solution? Does peat contribute that high (30%of OA) in previous studies? Usually, BBOA accounts for that high percentages.

Line 290: Please define R2.

Line 315: The percentage of OA to PM1 is not consistent to the rest of the text.

Section 4. The authors repeat the results as in the abstract and main results with no clear conclusion and "take home message" made by this work.

Tables 1 and 2. NOx should be mentioned in ppb.

Figure 1. Figure should be replaced by Fig. S1 (without Rathimnes station). Fig. 1c

seems to be wrong in comparison to Fig. S1 that includes all the measured data. Labels of elements seem wrong placed, making them hard to be read.

Figure 2. All chart pies should have the same size, the (a), (b), (c), (d) labels should be re-arranged accordingly to the main text. SO4 should be written as SO4 -2. The same applies for NH4, and NO3 (for the entire manuscript).

Figure 3. The (a), (b), (c), (d) should be re-arranged. Current Fig. 3b should go up to 5 $\mu$g m-3, so that the trend of PM1 components will be visible. SO4, NH4 and NO3 should be changed according to the previous comment for Fig. 2.

Figure 4. Pie charts should have all the same size. Labels of factors should have some space between each other. The (a), (b), (c), (d) should be re-arranged accordingly to the main text.

Figure 5. The values seem like the 3rd quartile of the data and not the actual average values. Please replace the whole figure according to the above suggestions.

Figure 7. Define SEAI, SCO. Does this figure correspond to data related to your measurements? Labels of sources should have some space between each other.

Major Comments:

Having read the paper I am still not sure what the authors want a reader to learn.

To my understanding, the authors want to empathize to traffic and vehicular emissions and more precisely to the importance of the diesel ones in the kerbside site. The authors claim that "in residential site, traffic emissions were found to have a minor impact on air quality". Yet, the HOA contribution to OA is the same (18%) for kerbside and residential sites for P2 period, with its mass average concentration to be 1.2 $\mu$g m-3 for kerbside and 1.6 $\mu$g m-3 for residential, something in contrast to their final statement.

The possession of the aethalometer AE-33 at the kerbside site is not taken into advantage by the authors. How much of the measured BC is related to traffic and how much to biomass burning? This could help interpreting better the results of the PMF but also the HOA/BC ratio in section 3.5. Have the authors tried to split BC according to its origins and then conclude to resemblance or not compared to other studies?

Peat, coal and wood factors play a major role (33-50% of OA) during P2 in both sites, yet the authors do not discuss their significance or any related mechanisms. The authors should pay attention to combustion sources, since they seem to be of greater importance than traffic.

Also, the PMF solutions for P2 are too constrained. Have the authors tried to constrain less the solution? What are the results?

How oxidized the OA factors are? Could the authors calculate the O:C ratio of each factor? This would be really helpful to the scientific community.

---

## Author Response (AR1)

We are grateful to the referees for their insightful comments which helped to improve the manuscript. We provided point-by-point responses to the referee's comments below where our responses are in blue.

Referee #1

In this study, Lin et al. evaluated the chemical composition and organic aerosol (OA) sources of PM1 monitored simultaneously at a kerbside site and a residential site in Dublin during both non-heating and heating periods. The authors found that vehicle emissions were associated with the significant kerbside increment of black carbon during the non-heating period, but they had a minor impact on air quality at the residential site. Significant contributions of solid fuel burning to OA were observed at both sites during the heating period. The findings in this study provided valuable information for aerosol scientists to better understand the temporal and spatial variations of the concentration, chemical composition, and OA origins of PM1 in Ireland, which could help the formulation of air quality policies and PM1 mitigation strategies in Ireland. Moreover, the paper is well written, and the results are visualized in an appropriate way. Therefore, I recommend it for publication after minor revisions.

Response: We thank the referee for the positive feedback.

Specific comments: 1. Line 14, ": : :sources of submicron aerosols (PM1): : :" Actually, the authors performed the source analysis of OA instead of PM1 in this study. Please revise.

Response: Corrected.

2. I think it is more straightforward to use "the non-heating period" and "the heating period" instead of "P1" and "P2" in the manuscript, especially in the figures and tables.

Response: As suggested by the referee, changes have been made to the manuscript, as well as in the figures and tables.

3. Line 149 and 170, the spikes of BC and OA were defined as those with measured concentrations higher than 15 $\mu$g/m3 and 5 $\mu$g/m3, respectively. What are the criteria to set these two threshold values?

Response: 15 $\mu$g m$^{-3}$ and 5 $\mu$g m$^{-3}$ are roughly the top 5 percentiles of BC and OA, respectively, during the non-heating period. We have clarified this in the revised manuscript.

4. Line 186. As mentioned by the authors, the dominant wind direction during the non-heating period was southwest while the city center was located to the north of the residential site. Was there any connecting flow between these two sites? If so, how did the BC concentration vary at the two sites under the connecting flows?

Response: Figure R1 shows the wind speed and wind direction frequencies during the entire sampling period. Southwesterly wind dominated over the entire period with over 50% of occurring frequency while north-northwesterly winds had a frequency of ~20% (Figure R1). The sampling site the kerbside is to the north of the residential site. When north-northwesterly wind prevailed, the median BC concentration at the residential site was 1 $\mu$g m$^{-3}$ in the morning while it was 5 $\mu$g m$^{-3}$ at the kerbside (Figure R2). Thus, the morning rush hour peak of BC was 80% reduced at the residential site when compared with kerbside.

35    Similarly reduced mixing ratios of NOx were also observed at the residential site (Figure R3). However, for the evening BC peaks occurring at 20:00 – 22:00 (local time), the median BC (3-4 µg m$^{-3}$) and 75$^{th}$ percentile (6-10 µg m$^{-3}$) were comparable between the two sites (Figure R2), primarily due to the emissions from residential solid fuel burning. These comments have been added to the revised manuscript/supplementary.

[Figure]

**Frequency of counts by wind direction (%)**

40              Figure R1. Wind speed and wind direction frequencies during the entire sampling period.

[Figure]

Figure R2. Diurnal patterns of BC measured at the kerbside site (BC_Kerb) and BC at the residential site (BC_Res) under the same wind directions. Solid lines represented median value while shade areas represent 75$^{th}$ and 95$^{th}$ percentiles.

[Figure]

Figure R3. Diurnal patterns of NOx measured at the kerbside site (NO$_x$_Kerb) and NO$_x$ at the residential site (NOx_Res) under different wind directions. Solid lines represented median value while shade areas represent 75$^{th}$ and 95$^{th}$ percentiles.

5. Line 194, Fig. 1c and 1d, another peak concentration was shown in the evening on 29 October, 2018. The authors suggested that these should be due to the residential heating activities. Their impacts on PM1 at residential site are comparable on both dates. However, the burning activity on 31 October showed a greater impact on PM1 than that on the other date at kerbside site. Do the authors have any idea what these burning activities are? Do these kinds of burning activities usually happen during the heating period in Ireland or just special for this sampling period? Are they open-field biomass burning? The reason for me to ask about this is obviously these two burning activities have pulled up the mean values of PM1, OA, BC, NOx etc. at the two sites during the heating period. More information about these burning activities will be helpful for the data interpretation and discussion in line 243-250.

Response: Original data in Fig. 1c was with 5-min resolution while Fig. 1d was 1 h data. As also pointed out by the second referee, we now present hourly averaged data for better comparison. As shown in Fig. R4, at the kerbside, the hourly averaged PM$_1$ peak on 31 October (134.0 µg m$^{-3}$) was substantially greater than on 28 October (85.0 µg m$^{-3}$) while the PM1 peaks were comparable at both dates (126.0 vs. 113.1 µg m$^{-3}$) for the residential site (Figure R4). This is because the pollution level at the kerbside was already elevated due to traffic emissions on 31 October (Figure R4) while the traffic pollution level on 28 October was less significant due to the meteorological conditions (e.g., higher wind speeds as shown in Figure S4). Therefore, heating emissions were added on top of the traffic emissions at the kerbside on 31 October, resulting in higher PM1 concentrations at the kerbside.

These kinds of burning activities from residential heating happen frequently in winter Ireland. In addition to the events reported in this study (winter 2018), our previous studies conducted in winter 2016 at the same residential site have also reported substantial contribution from heating sources (Lin et al., 2018; Lin et al., 2019b). In this study, we extended our measurement to the kerbside and the comparison of simultaneous measurement between the kerbside and residential site

improved our understanding of the spatial variation of aerosols and impacts of traffic/heating emissions. These discussions and relevant references have been added to Sect 3.4 in the revised manuscript.

[Figure]

Figure R4. Time series of the species measured by ACSM and AE-33/16 at the kerbside and residential site during the heating period in October 2018. The red line shows the trend of traffic emissions without the influence of heating while the blue cycle highlights the periods with aerosol increment due to heating emissions.

70

6. Line 234, in addition to the low temperatures in the evening, the PBL during the evening was usually lower than that during the day, and thus PBL could also partly contribute to the higher concentration of COA.

Response: Corrected. It now reads, "…likely due to higher emissions during the evening coupled with relatively low evening temperatures (Fig. S4) and shallower boundary layer."

75

7. Line 241, could the authors estimate the fraction of OOA that were regionally transported to the kerbside site?

Response: The diurnal trend of OOA (Fig. 4 in the revised manuscript) suggests the background OOA concentration was 0.5 $\mu g\ m^{-3}$ while the average OA concentration over the same period was 1.3 $\mu g\ m^{-3}$. Therefore, approximately 38% of the resolved OOA was regionally transported. In the revised text, it now reads, "The diurnal trend of OOA suggests the OOA had a background concentration of 0.5 $\mu g\ m^{-3}$ which was higher than HOA (0.1 $\mu g\ m^{-3}$) and COA (0.1 $\mu g\ m^{-3}$), indicating part of

80

85    OOA was also associated with regional transport. It is estimated that approximately 38% of the total OOA was regionally transported, i.e., the value of background OOA concentration of 0.5 µg m$^{-3}$ if compared to the total OOA concentration of 1.3 µg m$^{-3}$."

8. Line 296, it is not clear what are the typical values of HOA/BC for the gasoline vehicles-dominated environment. Please
90    add relevant values and references.

Response: HOA/BC ratios in the range of 0.9-1.7 have been reported in the gasoline vehicles-dominated environment (DeWitt et al., 2015). We have added such values and references in the revised manuscript. It now reads, "Compared to other studies, the HOA/BC ratio in Dublin was lower than the HOA/BC ratios reported for gasoline vehicles-dominated environment (0.9-1.7) but was within the range for the diesel environment (0.03-0.61) (DeWitt et al., 2015), indicating most of the traffic
95    emissions were from diesel vehicles."

9. Was the second ACSM also equipped with a quadrupole mass spectrometer? How were the Q-ACSMs calibrated?

Response: The second ACSM was also a quadrupole ACSM (i.e., Q-ACSM), we have clarified this in the revised Methods Section.
100      Both ACSMs were calibrated with ammonium nitrate and ammonium sulfate following the procedure described by Ng et al. (2011b). Briefly, monodispersed 300 nm ammonium nitrate/ammonium sulfate particles were generated by an atomizer (Model 9302, TSI) and size-selected by a differential mobility analyzer (DMA; TSI model 3080), and subsequently introduced into the Q-ACSMs to determine the response factor (RF), as well as the relative ionization efficiencies (RIEs) of ammonium and sulfate. We have added the above information in Sect. 2.2 in the revised manuscript.
105

10. Tables 1 & 2 can be combined into one. Please add PM1 data in the table. Moreover, numbers in the Table should have the same significant figures as those in the main text of the manuscript and figures.

Response: As suggested by the referee we have combined Tables 1 & 2 into one, have added PM1 data, and adjusted the significant figures in the revised table.
110

11. Several typos: 11. Fig. 4d, the OA concentration should be 8.1 instead of 8.7 $\mu$g/m3.

Response: The original Fig. 4d (i.e., pie charts) has now been combined to Fig. 2 in the revised manuscript. We provide the value of OA concentration in Table 1.

115    12. Line 43, "micron-environment" should be "micro-environment".

Response: corrected.

13. Line 133, should be Crippa et al. (2013).

Response: corrected.

120

Referee #2

Lin et al. (2020) in this study measured the chemical composition of submicron aerosol (PM1) in two locations in Dublin (kerbside and residential sites) from 4 September to 9 November 2018. Measurements were performed by using an Aerodyne Aerosol Chemical Speciation Monitor (ACSM) and an Aethalometer (AE-16 and AE-33). PMF analysis was also implemented

125  on the data. In general, the manuscript is well written though the results presented in the main manuscript do not correspond to the entire period of 4 September to 9 November 2018, but to only two weeks of measurements. The authors chose two periods of the available data, with the first one corresponding to a rather clean period (P1) with low mass concentration of PM1 and another one rather polluted (P2) by local sources (heating purposes). The study presented is very rich in data and analysis. However, these information are not very clear in the main manuscript or not even included (they are left on the

130  supplementary) making the manuscript weak and less readable, especially in the results section. Additionally, the reading might be challenging as OA sources and changes in concentrations are vaguely discussed and not really explained. Moreover, figures need reshaping as labels are not well positioned or data are not clear due to high values of the axis.

Response: We have included more information about the data and their analysis, and have reshaped figures substantially in the main manuscript. In particular, we have added more discussion on OA sources in the result section.

135

Specific comments: Line 26: "solid fuel burning..." it should be explained for which period is referred to. Solid fuel burning was not present in both periods.

Response: Clarified. Now it reads, "…solid fuel burning (38% or 2.4 µg m$^{-3}$, resolved only during the heating period)…"

140  Line 77: the word "monitor" is missing.

Response: Corrected.

Line 107: The 1 hour interval that was set for ACSM in the residential site is really high. This might lead to the lack of data originating from local sources and may affect the corresponding diurnal profiles of OA sources. Authors chose this in order to

145  reduce uncertainty though in Fig. S1b the measured PM1 mass concentration is quite high reaching values greater than the Kerbside site.

Response: We do not expect high variability in local sources at the residential site. In particular, our previous studies (30 min resolution) at the same residential site (Lin et al., 2018) shows the measured PM$_1$ was in good agreement with the collocated SMPS measurement with 15 min, as well as PM$_{2.5}$ (1 h) at another residential site (i.e., Rathmines station). Moreover, the

150  source apportionment of OA at the residential site suggested the major local sources were domestic solid fuel burning which was consistent with our previous study at the same site (Lin et al., 2018; Lin et al., 2019b). Therefore, we believe the hourly

resolution had captured the major local source at the residential site. Future studies using HR-ToF-AMS are to be conducted to study other local emissions at the residential site with higher time resolution (e.g., 5 min).

As the reviewer said, we had high concentrations during the heating period at the residential site. But for most of the time, especially at the beginning of the campaign, the $PM_1$ concentrations were relatively low (<2 μg m$^{-3}$). In the revised manuscript, it now reads, "a 1-hour interval was used at the residential site to reduce uncertainty in measurement due to the relatively low concentrations for most of the time"

Line 110: How much the collection efficiency was on average?

Response: The average collection efficiency was 0.5 on average. We have added this information to the revised Method Section.

Line 111: The Rathmines site is mentioned. It was not described before in the manuscript and is referred also afterwards without any information. Please provide some info.

Response: We have provided relevant information regarding the Rathmines site. It now reads, "Chemical compositions of $PM_1$ were simultaneously measured at a kerbside site in Dublin city centre and a residential site in suburban Dublin, at a distance of ~5 km from the kerbside site (see the map in Fig. S1) while $PM_{2.5}$ was sampled at Rathmines (https://www.epa.ie/air/quality/data/rm/), another residential site at a distance of ~3 from the kerbside and residential sites." Also, reads "… (i.e., Rathmines, Fig. S1) where a tapered element oscillating microbalance (ThermoFisher Scientific) was deployed with 1 h resolution".

Line 118: CO measurements are available for kerbside but they are not presented anywhere in the main manuscript.

Response: We have added a discussion on CO as well as NOx in the main manuscript. It now reads, "Specifically, both NOx and CO showed rush hour peaks during weekdays while such rush hour peaks during weekends were not as prominent as during weekends, consistent with the traffic pattern in Dublin (Fu et al., 2017). The average mixing ratios for NOx and CO were 54.1 ppb (in the range of 5.4-200.0 ppb) and 0.2 ppm (in the range of 0.05-0.8 ppm), respectively."

Line 137: Figure S1 deserves to be in the main manuscript. The overall (2 months period) chemical composition of submicron aerosol (PM1) should also be included for both sites and discussed.

Response: Figure S1 (or Figure R5) has now been moved to the main manuscript with a discussion of the overall chemical compositions of $PM_1$ for the entire campaign. It now reads, "Figure 1 shows the time series of submicron organic aerosol (OA), sulfate, nitrate, ammonium, chloride, and black carbon (BC) at both the kerbside and the residential site, as well as the time series of $PM_{2.5}$ at Rathmines from 4 September to 9 November 2018. The air quality during the early stage of the sampling period (before 30 September) showed limited influence from residential heating emissions as observed by few pollution spikes in the evening when compared to a later period with prominent evening spikes observed at all three sites (From 1 October to 09 November; Fig. 1). The three sampling sites (i.e., kerbside, residential, and Rathmines) are within a 5 km radius in Dublin

city and the time series of PM$_1$ and PM$_{2.5}$ were well correlated with linear correlation coefficients determination (R$^2$) in the range of 0.56-0.83 and slopes of 0.72-0.88 (Fig. S1). The good time series correlation between the three sites was mainly driven by the pollution events during the heating period while the slopes of 0.72-0.88 suggested PM$_1$ explained, on average, 72-88% of PM$_{2.5}$ mass. The poorer correlation (R$^2$ of 0.56; Fig. S1) between the kerbside PM$_1$ and Rathmines PM$_{2.5}$ than between the residential PM$_1$ and Rathmines PM$_{2.5}$ (R$^2$ of 0.83) was due to traffic emissions which had a greater impact on the kerbside than at both the residential site and Rathmines.

Over the entire period, the mean mass concentration of PM$_1$ was 11.7 $\pm$ 9.7 (one standard deviation) µg m$^{-3}$ at the kerbside with black carbon accounting for 50% of the total PM$_1$ mass (Fig. 1a), followed by OA (31%), nitrate (6%), sulfate (5%), ammonium (5%), and chloride (2%). At the residential site, the mean PM$_1$ concentration (6.6 $\pm$ 3.1 µg m$^{-3}$) was roughly half of that at the Kerbside (Fig. 1b). However, the chemical compositions of PM$_1$ at the residential site was dominated by OA (59% of PM$_1$), followed by BC (13%). The total inorganic aerosols (sum of sulfate, nitrate, ammonium, and chloride) accounted for 28% of PM$_1$ at the residential site, slightly higher than at the kerbside (18%), though the concentrations were comparable (1.8 vs. 2.1 µg m$^{-3}$) between the two sites, suggesting regional sources/formations rather than traffic emissions. In contrast, the greater abundance of BC at the kerbside suggests traffic emissions comprised a high fraction of BC which, however, had a limited impact on the residential site. In particular, BC at the kerbside showed two rush hour peaks, confirming traffic was its major source (Fig. S2). However, the BC was up to 25 times lower at the residential site during the rush hours (Fig. S2), suggesting a minor impact of traffic on the air quality at the residential site due to the effects of the wind direction and the distance from the road. In addition to the rush hour peaks of BC, an additional BC peak in the evening was also observed at both sites due to the emissions from domestic heating activities (discussed later). "

[Figure]

Figure R5 (or Figure 1 in the revised manuscript). Time series of hourly averaged submicron organic aerosol (OA), sulfate ($SO_4^{2-}$), nitrate ($NO_3^-$), ammonium ($NH_4^+$), chloride ($Chl^-$), and black carbon (BC) at the kerbside (a), residential site (b), and EPA Rathmines station (c) from 4 September to 9 November 2018. Inset pie charts are the chemical composition of $PM_1$ averaged over the entire period while the numbers (±one SD) above are the average $PM_1$ concentration. The non-heating period, from 10 September to 17 September, and heating period, from 27 October to 4 November 2018 are marked for further analysis. The data gaps at the kerbside were due to ACSM malfunction.

[Figure]

Figure R6 (or Figure S1). (a) Sampling sites for $PM_1$ measurements in Dublin marked by the red cycles at the kerbside (i.e., Trinity College Dublin) and residential site (i.e., University College Dublin), as well as for $PM_{2.5}$ measurement at Rathmines; (b) time series of $PM_1$ and $PM_{2.5}$; (c) scatter plot between residential $PM_1$ and Rathmines $PM_{2.5}$; and (d) scatter plot between Kerbside $PM_1$ and Rathmines $PM_{2.5}$. Also shown in (c) and (d) are the slopes and correlation coefficients ($R^2$) from the linear relationship. Note that removing the outlier in (c) resulted in a slope of 0.93 instead of 0.88. The map was adapted from Google Maps.

Line 151: Please specify Fig.1 and Fig.2 with Fig.1a, Fig.2a.

Response: Corrected.

Line 168: Fig.3a is mentioned before Fig. 1b, c,d and Fig. 2b, c, d. Please fix this.

Response: We have now combined the original Fig. 1 and Fig. 2 into one figure (now Fig. 2) in the revised manuscript. For
better comparison and visualization of the data, we tend to put the same contents (e.g, time series, diurnal) sampled at the two
sites in one Figure (i.e., Fig. 2 and Fig. 3). But in Sect. Results and Discussion, we tend to discuss e.g., time series and diurnal
of $PM_1$ together for the same site.

Line 173: "(Fig. 1)" Please define to which letter of Fig. 1 the text is referring to.
Response: Corrected.

Line 195: "197.3 $\mu$g m-3" This value is not consistent for Fig.1 c and Fig. S1a. Please provide the correct figure/ value.
Response: Original Fig. 1c was based on 5-min resolution data while Fig. S1a was based on hourly averaged data. To be
consistent and for better comparison, all data are now hourly averaged in the revised manuscript.

Line 207: Nitrate can be also present in higher values due to organonitrates (ON). Have the authors checked the fraction of
ON to nitrate?
Response: Organo-nitrates can be well estimated by HR-ToF-AMS based on m/z 30 and 46 (Kiendler-Scharr et al., 2016).
However, for UMR Q-ACSM, ambiguity remains due to the insufficient mass resolution to distinguish the interference of
$CH_2O^+$ from $NO_2^+$ at m/z 30. For such reasons, we have not investigated the fraction of ON to nitrate.

Line 215: "(Fig. S1)" should change to Fig. S1c
Response: Corrected.

Line 226: Fig. 5a is referred before Fig. 4b, c, d.
Response: Corrected.

Line 234: Higher COA mass concentrations could be related to the decrease of boundary layer.
Response: Corrected. Now it reads, "Higher dinner time COA peak (2.7 $\mu$g m$^{-3}$) was observed than lunch time peak (1.6 $\mu$g
m$^{-3}$) likely due to higher emissions during the evening coupled with relatively low evening temperatures (Fig. S2) and
shallower boundary layer."

Line 235: Fig. S7 does not correspond to the OOA profile spectrum.
Response: Corrected.

Fig. S6a, b deserves to be in the main manuscript instead of Fig. 5a.
Response: The updated Fig. S6 has been added to the main manuscript.

[Figure]

Figure R7 (or Figure 4 in the text) (a) Mass spectral profiles of hydrocarbon-like OA (HOA), cooking OA (COA) and oxygenated OA (OOA) during non-heating at the kerbside; (b) Diurnal cycle of the OA factors; and (c) relative contribution of OA factors over the entire non-heating period; (d) over the morning rush hours (7:00-10:00, local time); and (e) over the evening hours from 19:00-22:00. Error bars in (a) and (b) represent one standard deviation.

Line 235: The authors state that OOA resembles of the less volatile OOA (LV-OOA). What criteria lead to this statement?

Response: This was based on the fact that the correlation coefficient ($R^2$) between the resolved OOA profile and the reference profile of LV-OOA from Ng et al. (2011a) was 0.94 compared to 0.4 between OOA and SV-OOA (Ng et al., 2011a). We have added this point to the revised manuscript.

Line 243: The authors have constrained with ME-2 tool, 4 out of 5 factors with a relative strict $\alpha$-value. This leads to two factors (wood and coal) contributing fairly low to OA (4% and 8%, respectively), making the results less robust. How confident are the authors about this constraint? It seems that solution was forced to an erroneous solution, comparing the data to the unconstrained ones, where biomass burning factor contributed for approx..20% of OA.

Response: The reason we constrained 4 factors (i.e., HOA (or referred as oil), peat, coal, and wood OA factors) was to evaluate their contribution to ambient OA during the heating period. This was based on priori information that oil, peat, coal, and wood burning were used for heating according to the Central Statistics Office (CSO, 2016) and Sustainable Energy Authority of Ireland (SEAI, 2018). Previous offline (i.e., filter-based) and online (i.e., ACSM/AMS) studies in Ireland winter have identified

ambient OA being associated with the residential burning of oil, peat, coal, and wood burning (Kourtchev et al., 2011; Dall'Osto et al., 2013; Lin et al., 2018). In particular, our previous study has characterized the mass spectral profiles of peat, coal, and wood burning OA factors by simulating their burning in a typical Irish stove with an ACSM (Lin et al., 2017). Peat and wood burning both show biomass features with a prominent contribution at m/z 60 (i.e., f60), a marker fragment for biomass burning (Lin et al., 2017). Therefore, both peat and wood burning have contributed to the BBOA factor as resolved from the free PMF analysis. On the other hand, the increase in the concentration of chloride in the evening suggested coal use for heating, consistent with our previous study (Lin et al., 2017).

However, as pointed out by the referee, large uncertainties might exist with a tight constraint (i.e., a value of 0.1) when constraining 4 out of 5 factors. Therefore, we have conducted a sensitivity test by varying the *a* values of 0-0.5 (or 0-50% variation) for the reference profiles. In addition, a bootstrap-based resampling strategy with a total of 500 runs was applied to examine the statistical uncertainty using SoFi (Version 6.A1; https://datalystica.com/sofi). The following criteria are used for the selection from these ME-2 runs to get the most optimized solution:

1, Correlation between the time series of HOA and $NO_x/BC$ in the morning (7:00-10:00) during weekdays. HOA and $NO_x/BC$ have a common source from traffic emissions in these morning hours.

2, Diurnal cycle of COA. The COA concentrations during mealtime should be higher. In this study, the ratio of COA at 13:00 to the average of COA of 10 and 11 am was monitored.

3, Multi-linear regression between BC and heating-related factors (i.e., HOA (or oil burning), peat, wood, and coal burning) in the evening (19:00-23:00), assuming only the heating-related factors contributed to BC in the evening.

4, Fraction of m/z 60 (i.e., f60) for peat and wood burning OA factors should be higher than 0.006 (Cubison et al., 2011; Lin et al., 2017).

5, Fraction of m/z 44 (i.e., f44) for the unconstrained factor (i.e., OOA). OOA should have higher f44 than primary factors.

Final time series and mass spectra were the averages of the ME-2 runs meeting these criteria. In the revised manuscript, we have clarified these new ME-2 strategies, and have provided the new results.

Line 244: Fig. 4c and Fig. 5a are referred before Fig. 4b and Fig. 5b

Response: Corrected.

Line 245: There is no such high peak in HOA around 23:00, according to Fig. S11b. It seems that on Fig. 5 the 3rd quartile of the data has been used instead of the average values. This is consistent to the average values given in the manuscript. Again, Fig. S11 deserves to be in the main text instead of Fig. 5c.

Response: The original Fig. S11b shows the median and 25th and 75th quartiles while the original Fig. 5c shows the average values of the OA factors stacked on top of each other. To be consistent, we have used the average values in both the main text and supplementary.

As suggested by the referee, we have moved the updated Fig. S11 in the revised manuscript (see the figure below).

[Figure]

Figure R8 (or Figure 6 in the text). (a) Mass spectral profiles of hydrocarbon-like OA (HOA), cooking OA (COA), peat, wood, coal and oxygenated OA (OOA) factors during heating at the kerbside; (b) Diurnal cycle of the OA factors; and (c) relative contribution of OA factors over the entire heating period; and (d) over evening spikes on 28 and 31 October. Error bars in (a) and (b) represent one standard deviation.

Line 253: peat, coal and wood factors contribute one third to the total OA during P2 in kerbside. However, only a line is describing their existence.

Response: We have now added more discussion in the revised manuscript. It now reads, "Peat is an accumulation of partially decayed plant material which is an important domestic fuel source in Ireland (Tuohy et al., 2009). The incomplete decay of vegetation resulted in an increase of $f60$ when burned (Lin et al., 2017). However, $f60$ in peat profile (0.014) was lower than wood (0.053) probably because wood contained a higher fraction of $m/z$ 60-related material e.g., levoglucosan (Fig. 6a). Over the entire heating period, peat burning accounted for 17.7% (1.2 µg m$^{-3}$) of OA (Fig. 6c). However, during the pollution spikes as seen on 28 and 31 October (Fig. 6d), peat burning increased its fraction to 26.5% (9.2 µg m$^{-3}$), suggesting peat burning was an important OA emission source. Similarly, wood burning increased its contribution to 9.8% (3.3 µg m$^{-3}$) during pollution

spikes from an average of 7.3% (0.5 μg m$^{-3}$). The important role of peat and wood burning in driving the pollution events is consistent with our previous study in suburban Dublin (Lin et al., 2018).

The profile of coal burning OA featured very low *f60* (<0.003), consistent with its non-biomass signature as coal is formed from the complete vegetation decay. On average, the coal factor accounted for 12.6% (0.8 μg m$^{-3}$) of OA. Though the fraction of coal burning decreased during pollution events (11.9%; Fig. 6d), its absolute concentration increased (4.3 μg m$^{-3}$). Note that chloride also showed a significant increase during the pollution events (Fig. 2c) which was associated with coal burning as our previous coal-combustion experiment showed that chloride emission comprised a high fraction (2-52.8%) of the submicron aerosol from coal burning emissions (Lin et al., 2017). "

Line 254-255: Could this higher OOA levels be transported by another area and not due to SVOCs?

Response: Compared to the non-heating period, OOA was significantly elevated, coincided with the increase of primary OA factors, in the evening during the heating period. Thus, it is likely that higher OOA levels in the evening during heating were due to the secondary formation/aging of pollution plumes from heating sources. In the revised manuscript, it now reads, "The OOA profile had an *f44* of 0.24 during heating which was similar to that (0.22) during non-heating, indicating similar oxidation levels between the two periods. However, compared to the daytime OOA peak concentrations (1.8 μg m$^{-3}$), higher OOA concentrations (3.9 μg m$^{-3}$) were observed in the evening, indicating a more important contribution from the condensation of SVOCs emitted from heating sources and/or their dark aging processes (Tiitta et al., 2016)."

Section 3.3.2: Lacks of discussion.

Response: We have now added more discussion on the OA sources at the residential site during non-heating (Sect. 3.4.2) and heating (Sect. 3.4.3) period.

In Sect. 3.4.2., it now reads, "Free PMF suggested a two-factor solution with one mixed primary OA factor and one OOA (Fig. S8) as a further increase in the number of factors led to the splitting of factors. The diurnal pattern of the primary factor showed a slight increase in the morning but with a larger increase in the evening and night, suggesting potential mixing between HOA and the heating-related factor. Note that the temperature was below 15 $^{o}$C in the evening during this period (Fig. S4) and thus sporadic domestic solid fuel burning activities were likely to occur. Our previous study has shown that peat burning occurred in cold summer nights in the west coast city of Galway in Ireland (Lin et al., 2019a). Moreover, the elevated levels of m/z 60 in the evening (at 22:00) suggested emissions from biomass burning (Fig. S8). The correlation between the profile of the unconstrained primary OA factor and the reference profile of peat burning OA (R$^2$= 0.58) from our previous study (Lin et al., 2017) is better than the profile of biomass burning OA (BBOA; R$^2$=0.44) from the Ng et al. (2011a). Note that COA was not considered to be a potential OA factor at this location since the sampling site was representative of the residential area with few restaurants around. Moreover, no clear increase in the concentration of POA during lunchtime was observed during this non-heating period and the heating period as discussed later. Therefore, only the reference profile of HOA and peat burning OA factors were constrained during non-heating at the residential site.

360       The mass spectra and diurnal patterns, as well as the relative contribution of the HOA, peat, and OOA at the residential site during non-heating are shown in Fig. 5. While the profile of HOA is similar between the residential site and the kerbside, its concentration levels at the residential site were significantly lower than at the kerbside. Specifically, in the morning rush hours, the HOA peak concentration was 0.1 µg m$^{-3}$ at the residential site while it was 1.5 µg m$^{-3}$ at the kerbside, with a 15 times difference between the two sites. On average, HOA accounted for 11.8% of OA at the residential site. During the morning

365 rush hours, its fraction increased to 16.0% which was still a minor factor when compared to OOA (58-68% of OA). The low contribution of HOA is consistent with the low mixing ratio of NO$_x$ at the residential site (median: 2.0 ppb) which was 20 times lower than that at the kerbside (Table 1).

     The peat profile featured peaks at *m/z* of 27, 29, 41, 43, 55, and 57, which was similar to HOA. However, the differences in *f60* between the peat factor and HOA suggested different sources. *f60* in the peat profile was 0.014 which was higher than that

370 for HOA (<0.003), confirming its biomass nature (Cubison et al., 2011). The diurnal pattern of peat showed increased concentrations at 20:00-22:00, corresponding to their emission time. On average, peat accounted for 23.6% of OA and its fraction increased to 28.1% in the evening. The OOA mass spectral at the residential site featured high contribution at m/z 44 (i.e., *f44*) which was similar to the OOA at the kerbside. However, the diurnal cycle of OOA at the residential site showed no clear pattern, which was different from that at the kerbside. Therefore, most of the OOA at the residential site was likely due

375 to regional transport."

Also, in Sect. 3.4.4, it now reads, "Heating-related OA factors were identified since they all showed elevated concentrations in the evening as indicated by the free PMF solutions (Fig. S10). However, similar to the case at the kerbside, the OA factors were mixed because of the co-emissions from all domestic heating activities. To better evaluate the contribution of potential sources, the reference profiles of HOA (Crippa et al., 2013), peat, wood, and coal were constrained (Lin et al., 2017) using

380 ME-2. Note that COA was not constrained since no lunch meal peaks were identified during this period as discussed above.

In the morning rush hours, HOA showed a peak concentration of 0.4 µg m$^{-3}$ due to traffic emissions, which was, again, largely reduced when compared to the HOA morning peaks (1.8 µg m$^{-3}$) at the kerbside. However, higher concentrations of HOA (2.0 µg m$^{-3}$) were observed in the evening due to the emissions of oil burning. Therefore, the majority (estimated at over 80%) of HOA at the residential site was due to oil burning instead of traffic emissions. On average, HOA accounted for 11.8% of the

385 total OA over the entire heating period and its fraction increased slightly to 12.9% during pollution events.
 Similar to the kerbside, solid fuel burning, especially peat burning was a very important OA factor, contributing up to 45.5% (34.9% on average) of the OA during the pollution events (Fig. 7c and 7d). Coal (17.8-19.2%) and wood (7.1-7.5%) burning were also contributing substantially over the entire period as well as during pollution events. The importance of solid fuel burning at the residential site echoed our previous studies at the same sites (Lin et al., 2018; Lin et al., 2019b). OOA, on

390 average, accounted for 27.9% (1.4 µg m$^{-3}$) of OA and the higher OOA concentration during the evening, concurrent to an increase in primary factor concentrations, again suggested its major source was from the condensation of SVOCs emitted from solid fuel burning and/or their dark aging processes (Tiitta et al., 2016)."

Line 258: A peat and a HOA factor were constrained with ME-2. Why the authors chose a peat factor instead of a BBOA one
395 from literature? Does this change the results significantly? Also, a COA factor is not present. Have the authors tried to constrain it? In this case how much was its contribution to the OA? Again Fig. S14a,b should be included in the main text.

Response: The reason we chose to constrain peat factor, instead of BBOA from literature, is that the correlation between the profile of the unconstrained factor and reference peat OA profile ($R^2$=0.58) is better than BBOA ($R^2$=0.44). Moreover, the diurnal pattern of the primary factor from the fee PMF solution showed a slight increase in the morning but with a larger

400    increase in the evening and night, suggesting potential mixing between HOA and the heating-related factor. Note that the temperature was below 15 °C in the evening during this period and thus sporadic domestic solid fuel burning activities were likely to occur. Our previous study has shown that peat burning was likely to occur in cold summer nights in the west coast city of Galway in Ireland (Lin et al., 2019a). Moreover, the elevated levels of m/z 60 in the evening (at 22:00) suggested emissions from biomass burning (Fig. S8). The correlation between the profile of the unconstrained primary OA factor and

405    the reference profile of peat burning OA ($R^2$= 0.58) from our previous study (Lin et al., 2017) is better than the profile of biomass burning OA (BBOA; $R^2$=0.44) from the Ng et al. (2011a). Note that COA was not considered to be a potential OA factor at this site since the sampling site was representative of the residential area with few restaurants around. Moreover, no clear increase in the concentration of POA during lunchtime was observed during this non-heating period and the heating period. Therefore, only the reference profile of HOA and peat burning OA factors were constrained during non-heating at the

410    residential site. We have added the above comments in the revised manuscript.

     The updated Fig. S14a, b has now been moved to the main text (see the figure below).

[Figure]

Figure R9 (or Figure 5 in the text). (a) Mass spectral profiles of hydrocarbon-like OA (HOA), peat and oxygenated OA (OOA)

415    during non-heating at the residential site; (b) Diurnal cycle of the OA factors; and (c) relative contribution of OA factors over

the entire non-heating period; (d) over the morning rush hours (7:00-10:00); and (e) over the evening hours from 19:00-22:00. Error bars in (a) and (b) represent one standard deviation.

Line 265: Again 4 out of 5 factors have been constrained with ME-2 tool. The unconstrained solution of this period (P2) for the residential site is not given in the supplementary so no direct comparisons can be made. How confident are the authors about this solution? Does peat contribute that high (30% of OA) in previous studies? Usually, BBOA accounts for that high percentages.

Response: See the response to the previous comment regarding the constraining strategy.

We have now provided the unconstrained PMF solution for the residential site in the supplementary. In the free PMF solution, heating-related OA factors were identified since they all showed elevated concentrations in the evening. However, similar to the case at the kerbside, the OA factors were mixed because of the co-emissions from all domestic heating activities. To better evaluate the contribution of potential sources, the reference profiles of HOA (Crippa et al., 2013b), peat, wood, and coal were constrained (Lin et al., 2017b) using ME-2.

We have explored the solution space with a large range of a-values (0-0.5) and used the bootstrap resampling strategy to evaluate the statistical uncertainties. We are confident about this solution as the results from this study were consistent with previous studies (Kourtchev et al., 2011; Dall'Osto et al., 2013; Lin et al., 2017).

Peat is a type of biomass since it is an accumulation of partially decayed plant material which is an important domestic fuel source in Ireland (Tuohy et al., 2009). The incomplete decay of vegetation resulted in an increase of $f60$ when burned (Lin et al., 2017). However, $f60$ in peat profile (0.014) was lower than wood (0.053) probably because wood contained a higher fraction of $m/z$ 60-related material e.g., levoglucosan (Fig. 6a). Our previous study has shown that peat burning was contributing substantially in suburban Dublin (Lin et al., 2018; Lin et al., 2019b).

Line 290: Please define R2.

Response: Now defined as linear correlation coefficients determination ($R^2$)

Line 315: The percentage of OA to PM1 is not consistent to the rest of the text.

Response: Now changed to 46-64%, consistent with the rest of the text.

Section 4. The authors repeat the results as in the abstract and main results with no clear conclusion and "take home message" made by this work.

Response: The content in this Section has been incorporated into Section 3.1 when we discussed the overview of the measurement.

Tables 1 and 2. NOx should be mentioned in ppb.

 Response: corrected.

Figure 1. Figure should be replaced by Fig. S1 (without Rathimnes station).

Response: It is now replaced but we tend to keep the PM$_{2.5}$ data from Rathmines station since it provides a good comparison with our measurement. See Figure R5 in the previous reply.

455

Fig. 1c seems to be wrong in comparison to Fig. S1 that includes all the measured data. Labels of elements seem wrong placed, making them hard to be read.

Response: Corrected. And labels have been re-arranged.

[Figure]

460

Figure R10 (or Figure 2 in the text). Time series of hourly averaged submicron organic aerosol (OA), sulfate (SO$_4$), nitrate (NO$_3$), ammonium (NH$_4$), chloride (Chl), and black carbon (BC) during non-heating (left panel) and heating (right panel) at the kerbside (a, c) and residential (b, d). Inset pie charts are the relative fraction of the measured PM$_1$ components. Also shown is the day of the week, including Monday (Mon), Tuesday (Tue), Wednesday (Wed), Thursday (Thu), Friday (Fri), Saturday (Sat), and Sunday (Sun).

Figure 2. All chart pies should have the same size, the (a), (b), (c), (d) labels should be re-arranged accordingly to the main text. SO4 should be written as SO$_4^{2-}$. The same applies for NH4, and NO3 (for the entire manuscript).

470 Response: Pie charts now was combined with Fig. 2 with the same size in the revised manuscript (see Figure R10). Labels are re-arranged. And SO4, NH4, NO3, Chl have been changed.

Figure 3. The (a), (b), (c), (d) should be re-arranged. Current Fig. 3b should go up to 5 $\mu$g m-3, so that the trend of PM1 components will be visible. SO4, NH4 and NO3 should be changed according to the previous comment for Fig. 2.

475 Response: Changes made

[Figure]

Figure R11 (or Figure 3 in the text). The diurnal cycle of submicron organic aerosol (OA), sulfate ($SO_4^{2-}$), nitrate ($NO_3^-$), ammonium ($NH_4^+$), chloride ($Chl^-$), and black carbon (BC) during non-heating (left panel) and heating (right panel) at the kerbside (a, c) and residential site (b, d).

480

Figure 4. Pie charts should have all the same size. Labels of factors should have some space between each other. The (a), (b), (c), (d) should be re-arranged accordingly to the main text.

Response: Pie charts with same size are now combined to Figs. 4-7. Also see the response to the previous comment.

485 Figure 5. The values seem like the 3rd quartile of the data and not the actual average values. Please replace the whole figure according to the above suggestions.

Response: We have replaced the original figure with new figures from our new analysis. See the response to the previous comment.

490    Figure 7. Define SEAI, CSO. Does this figure correspond to data related to your measurements? Labels of sources should have some space between each other.

Response: SEAI is Sustainable Energy Authority of Ireland (SEAI, 2018) while CSO is Central Statistics Office (CSO, 2016), which is now added to the revised manuscript (Sect. Sources of OA). This figure was presented to show the energy structure regarding traffic and domestic fuels in Ireland and to investigate their possible relationship with the PM pollution. As pointed

495    out by the referee, this figure is not closely related to the measurement we had here. Therefore, we have moved this figure to the supplementary in the revised manuscript.

Major Comments: Having read the paper I am still not sure what the authors want a reader to learn. To my understanding, the authors want to empathize to traffic and vehicular emissions and more precisely to the importance of the diesel ones in the

500    kerbside site. The authors claim that "in residential site, traffic emissions were found to have a minor impact on air quality". Yet, the HOA contribution to OA is the same (18%) for kerbside and residential sites for P2 period, with its mass average concentration to be 1.2 $\mu$g m-3 for kerbside and 1.6 $\mu$g m-3 for residential, something in contrast to their final statement.

Response: In this study, we wanted to get insights into the spatial distribution of submicron aerosols in urban Dublin, particularly for traffic emissions. As has been discussed in the Sect. Introduction, our previous study conducted in suburban

505    Dublin has shown that air quality in Dublin was strongly influenced by the emissions from residential solid-fuel burning in winter (Lin et al., 2018). However, the impact of traffic emissions on this residential site was shown to be minor probably due to the distance (~500 m) from the nearest roads, as well as the strict emission standard (Lin et al., 2018). The minor influence of traffic at the residential site was also shown in a west coast city of Galway in Ireland during both summer (Lin et al., 2019a) and winter conditions (Lin et al., 2017). In particular, the diurnal pattern of HOA shows largely enhanced concentration in the

510    evening when compared to that during the morning rush hours, suggesting residential heating is the major source of HOA in suburban Dublin (Lin et al., 2019b). Heating source of HOA is also reported at the urban background site in Paris in addition to traffic (Petit et al., 2014; Zhang et al., 2019). However, the relative importance of traffic and heating to HOA in different urban settings (e.g., kerbside and residential) and different seasons (e.g., heating and non-heating) remain poorly understood.

HOA at the kerbside with rush hour peaks can be regarded as OA emissions from traffic emission as the case at the kerbside

515    during the non-heating period. However, during the heating period, the greater increase in HOA in the evening was associated with oil burning. Specifically, at the residential site, HOA showed a peak concentration of 0.4 µg m$^{-3}$ in the morning rush hours due to traffic emissions, which was largely reduced when compared to the HOA morning peaks (1.8 µg m$^{-3}$) at the kerbside. However, higher concentrations of HOA (2.0 µg m$^{-3}$) were observed in the evening due to the emissions of oil burning. Therefore, the majority (estimated at over 80%) of HOA at the residential site was from oil burning instead of traffic emissions.

520    We have added the above comments in the revised manuscript and highlighted it in the Abstract.

The possession of the aethalometer AE-33 at the kerbside site is not taken into advantage by the authors. How much of the measured BC is related to traffic and how much to biomass burning? This could help interpreting better the results of the PMF

but also the HOA/BC ratio in section 3.5. Have the authors tried to split BC according to its origins and then conclude to
525    resemblance or not compared to other studies?

Response: A 7-wavelength AE-33 was deployed at the kerbside along with an ACSM. During the non-heating period at the kerbside, BC shows two rush hour peaks, indicating traffic was its major source. Moreover, there was no evidence of biomass burning during non-heating at the kerbside as indicated by the PMF analysis. Therefore, BC during non-heating was exclusively from traffic emissions. The HOA/BC ratio derived from this period can better present the traffic HOA/BC ratio

530    since, during heating period, BC has additional sources from e.g., solid fuel burning. Figure R12 shows the HOA/BC ratio was 0.17 during the non-heating period at the kerbside which indicates a major source of diesel vehicular emissions (DeWitt et al., 2015).

During the heating period, an increase of BC concentration in the evening, with origins from heating (e.g., biomass/coal burning) sources, was also observed in addition to the two rush hour peaks. The Ångström exponent model can separate the

535    BC into traffic BC (BCtr) and wood burning BC (BCwb) (Sandradewi et al., 2008; Zotter et al., 2017). However, in Ireland, coal combustion and oil burning was also contributing the BC in addition to biomass burning and traffic. Thus, the Ångström exponent model could have large uncertainty to attribute the BC into just BCtr and BCwb. For such reasons, we have not included the BC source apportionment in the manuscript. Actually, we are preparing a separate study to address the BC sources.

[Figure]

540    Figure R12. Linear regression between HOA and BC with a slope of 0.17 and $R^2$ of 0.61 during the non-heating period at the kerbside.

Peat, coal and wood factors play a major role (33-50% of OA) during P2 in both sites, yet the authors do not discuss their significance or any related mechanisms. The authors should pay attention to combustion sources, since they seem to be of

545  greater importance than traffic.

Response: We have now discussed the significance of solid-fuel burning in the revised manuscript. Please find the response to the previous comment.

Also, the PMF solutions for P2 are too constrained. Have the authors tried to constrain less the solution? What are the results?

550  Response: We have now explored the solution space with less constraint (a value of 0-0.5). Please find the response to the previous comment.

How oxidized the OA factors are? Could the authors calculate the O:C ratio of each factor? This would be really helpful to the scientific community.

555  Response: Based on *f44*, we can estimate the O:C ratios for OA factors (Aiken et al., 2008; Canagaratna et al., 2015). Depending on the method used, O:C ratio was estimated to be in the range of 0.9-1.1 for OOA and 0.1-0.3 for primary OA factors. For OOA, the O:C ratio suggested OOA was highly processed. Moreover, the OOA profile resembled the less volatile OOA (LV-OOA; $R^2$ of 0.94) which usually represents well-aged SOA (Ng et al., 2011a). However, the diurnal pattern of OOA in Dublin showed a clear pattern that was strongly influenced by local sources (e.g., solid-fuel burning) and was most likely

560  from fresh SOA instead of well-aged SOA. Note that the unit mass resolution of ACSM makes it hard to separates the contribution between $C_2H_4O^+$ and $CO_2^+$ at m/z 44. Moreover, the "Pieber" effects need to be taken into account when calculating f44 (Pieber et al., 2016). Therefore, we tend not to report O:C ratios for OA factors due the large uncertainties.

[revised manuscript text omitted]

---

## Referee Report (RR1)

Specific comments:

Line 15: Space after (OA)

Line 24: Which is the corresponding ratio fpr the whole period?

Lines 25-27: Clarify the periods, heating – no heating.

Line 70: "Residential heating is the major source of HOA"

Line 84: 3 "km"?

Line 129: How far is the airport? Can we trust the meteorological data from that far away? How different is going to be the wind speed/ direction inside the city? Street effect doesn't affect these?

Lines 152-160: Use dots, instead of comas.

Line 153: $R^2$ for $NO_x$ and BC?

Line 185: Is there a wind dependency? When N/Ne winds BC was almost hlf, does wind affect the kerbside measurements?

Line 216: Mentioning the diurnal profile at this point can strengthen your point better that the 7-days diagram.

Line 234: I found it confusing, the discussion is about the kerbside measurements and the authors refer to Figure 2b which is about residential. Is this a typo?

Line 259: What about the boundary layer effect?

Line 271: Discuss a bit about the Chloride peaks on 31/10 to 1/11.

Comments about fig. 3c?

Line 282: Discuss a bit about the Chloride peaks on 31/10 to 1/11. Suggestions of origin?

Comments about Fig. 3d?

Line 290: Figure S7 and every OOA spectrum, the high m/z at the beginning in OOA is that of m/z 18? Why do you have such a big signal of water since a Nafion dryer was used?

Line 314: What LV-OOA, what is the source of that factor? Clarify better? Any other correlations to the rest of bibliography?

Line 321: How they get the 38% contribution? Was it compared to other pollutants? Please clarify.

Line 324: Why is there an evening peak?

Line 330: Maybe then this period was not appropriate to call it no heating?

Line 333: The authors state that they chose the peat factor because it had a better correlation than BBOA reported by Ng et al. (2011a). But there is no other comparison to the rest of BBOA factors in the rest of literature. Recent studies has shown that the spectra of BBOA factors can vary a lot, so taking into account only one publication may lead to incorrect results and conclusions.

Line 351: $f_{44}$ is present and common to every OOA factor. What similarity want the authors to show?

Line 352: It has a pattern. It peaks at night. Suggestions?

Line 397: What is the $R^2$ between the time series of the factors? Does the correlation indicate common source of the factors? (more factors than should be used etc?)

Line 412: This refer to Kerbside site?

Section 3.5. Discuss more. This is the main conclusion of this work and should include more information.

References: Should all have the same format.

Figure 1: Please redo it. Rathimnes is not very visible with this grey colour. Also, stucking components is creating problems like during the period of 4/10 to 23/10 at Kerbside site where it seems like there is no other component but BC.

Figure S3. Move it to main manuscript.

Figure 2a. During Sunday seems like there is only BC. Fix it.

Figure 2b. BC seems like contributing nothing. Fix it.

Figures 4, 5, 6, 7 (a). Fraction of what?

In overall the manuscript has been improved but still lacks in the section of scientific results.

In introduction the authors state that: "In particular, the diurnal pattern of HOA shows largely enhanced concentration in the evening when compared to that during the morning rush hours, suggesting residential heating is the major source of HOA in suburban Dublin. However, the relative importance of traffic and heating to HOA in different urban settings (e.g., kerbside and residential) and different seasons (e.g., heating and non-heating) remain poorly understood."

There is an effort in linking these statements to the current work but the authors should clarify better the relation between HOA and residential heating, something that is not very clear in the current version. The importance of HOA is not that high during the heating periods and maybe title should be reconsidered.

---

## Author Response (AR2)

We thank the referee for his/her careful reviewing the manuscript, which helped to improve it even further. Our point-by-point responses to the referee's comments are below in blue.

Specific comments:

Line 15: Space after (OA)

Response: corrected

Line 24: Which is the corresponding ratio for the whole period?

Response: For the whole period, the ratio was 7:1. But to be consistent with our overall concept and preceding statement (i.e., "Through the detailed comparison of one-week non-heating period from 10 to 17 September and one-week heating period from 27 October to 4 November…"), we tend not to report this value (i.e., the averaged value over the whole period) in the abstract. However, we have included it to the main text.

Lines 25-27: Clarify the periods, heating – no heating.

Response: We have now clarified the periods of heating (from 27 October to 4 November) and non-heating (from 10 to 17 September) in the revised manuscript.

Line 70: "Residential heating is the major source of HOA"

Response: Our previous study shows that the residential heating is the major source of HOA (Lin et al., 2018; Lin et al., 2019b). This is evidenced by the diurnal pattern of HOA which shows largely enhanced concentration in the evening when compared to that during the morning rush hours (Lin et al., 2019b).

Line 84: 3 "km"?

Response: corrected.

Line 129: How far is the airport? Can we trust the meteorological data from that far away? How different is going to be the wind speed/direction inside the city? Street effect doesn't affect these?

Response: The airport is about 5 km north to the sampling site at the kerbside. We do not expect big changes in wind speed/wind direction due to such distance, especially, when we want to study regional transport of aerosols and not local effects.

In the text, we have now specified the distance between the airport and the sampling site, it now reads "Meteorological variables (temperature, relative humidity, wind speed, and wind direction) with a time resolution of 1 hour were recorded at the meteorological stations (Irish meteorological service) of Dublin airport (~5 km north to the kerbside sampling site)."

Lines 152-160: Use dots, instead of comas.

Response: corrected.

Line 153: R2 for NOx and BC?

Response: $R^2 > 0.4$ for NOx and $R^2 > 0.6$ BC. The values were added to the revised text.

Line 185: Is there a wind dependency? When N/Ne winds BC was almost half, does wind affect the kerbside measurements?

Response: Yes, there is a wind dependency especially for the BC in the evening (i.e., residential BC). The BC in

N/Ne winds was not well represented in the current data because the frequency of N/Ne is lower (<5%) over the entire period as shown in the wind rose below (Fig. R1). In the revised Fig. S2, we have added the wind rose plot.

[Figure]

Figure R1. Wind frequency of counts by wind direction.

Line 216: Mentioning the diurnal profile at this point can strengthen your point better that the 7-days diagram.

Response: We have now mentioned the diurnal profile at this point in the revised manuscript. It reads, "As a result, the averaged BC diurnal profile showed both morning (9:00) and evening (17:00) rush-hour peaks (>10 µg m$^{-3}$; Fig. 4a)."

Line 234: I found it confusing, the discussion is about the kerbside measurements and the authors refer to Figure 2b which is about residential. Is this a typo?

Response: Sorry for the confusion. It was a typo, and we have now changed it to Figure 2a.

Line 259: What about the boundary layer effect?

Response: Boundary layer effect could also be partly contributing to the higher concentration in the evening, which has now been added to the revised manuscript. It now reads, "Note that the shallower boundary layer in the evening was also partly contributing to the elevated concentrations of the PM$_1$ components."

Line 271: Discuss a bit about the Chloride peaks on 31/10 to 1/11.

Comments about fig. 3c?

Response: We have now discussed a bit about chloride and Fig. 3c (now as Fig. 4c) in the revised manuscript. In the text, it now reads, "Moreover, the maximum chloride concentration (28.5 µg m$^{-3}$; Fig. 2c) was observed in the evening of 31 October along with other species, suggesting emission sources from solid fuel burning (Lin et al., 2017). While the diurnal pattern of PM$_1$ during the heating period still showed two rush hour peaks as found during the non-heating period (Fig.

4c), the higher PM$_1$ peak in the evening again highlighted the importance of the heating emissions."

Line 282: Discuss a bit about the Chloride peaks on 31/10 to 1/11. Suggestions of origin?

Comments about Fig. 3d?

Response: We have now discussed chloride peaks and suggested its origins. Comments about Fig. 3d (now as Fig.

4d) is also added in the revised manuscript. It now reads, "In particular, the maximum concentration of chloride (18.3 µg m$^{-3}$), observed in the evening of 31 October at the residential site, suggested emissions from solid fuel burning as found at the kerbside." Also, "Similar to that during the non-heating period, the averaged diurnal profile of the PM$_1$ components during the heating period showed a weak impact from traffic emission during the day (Fig. 4d). In contrast, the large increase of PM$_1$ components in the evening and night indicated substantial contribution from the heating sources."

Line 290: Figure S7 and every OOA spectrum, the high m/z at the beginning in OOA is that of m/z 18?

Why do you have such a big signal of water since a Nafion dryer was used?

Response: m/z 18 could also be from organic aerosol fragmentation in addition to water (Allan et al., 2004). In the fragmentation table used in ACSM or UMR AMS, m/z 18 was set to be equal to m/z 44 (Allan et al., 2004). Our

OOA mass spectrum is consistent with the profiles reported in the literature (Ng et al., 2011a; Fröhlich et al., 2015).

Line 314: What LV-OOA, what is the source of that factor? Clarify better? Any other correlations to the rest of bibliography?

Response: LV-OOA is low-volatility oxygenated OOA. LV-OOA is secondary organic aerosol which usually correlates well with nonvolatile secondary species such as sulfate (Jimenez et al., 2009). In this study, the resolved OOA correlated better with LV-OOA ($R^2$=0.94) than with SV-OOA ($R^2$=0.34). We have clarified this and added the correlation with SV-OOA as a comparison in the revised manuscript. It now reads, "The OOA profile (Fig. 5) resembles the low-volatility OOA (LV-OOA; $R^2$ of 0.94 between OOA and the reference LV-OOA profiles) (Ng et al., 2011a) which typically represents well-aged SOA and correlates better with non-volatile secondary species such as sulfate (Jimenez et al., 2009). As a comparison, the OOA profile is poorly correlated with semi-volatile OOA (SV-OOA; $R^2$ of 0.34 between OOA and the reference SV-OOA profiles) (Ng et al., 2011b)"

Line 321: How they get the 38% contribution? Was it compared to other pollutants? Please clarify.

Response: The background OOA was 0.5 µg m$^{-3}$ which was assumed to be the regional OOA. The background OOA (0.5 µg m$^{-3}$) was compared to total OOA (1.3 µg m$^{-3}$) to get the 38% contribution. We have clarified this in the original text, "It is estimated that approximately 38% of the total OOA was regionally transported, i.e., the value of background OOA concentration of 0.5 µg m$^{-3}$ if compared to the total OOA concentration of 1.3 µg m$^{-3}$."

Line 324: Why is there an evening peak?

Response: The evening peak is probably due to the condensation/fast processing of its precursor gases from e.g., cooking emissions. In fact, cooking emission has recently been found to be very important secondary aerosol sources in urban areas (Liu et al., 2018). We have discussed the possible source of the OOA peak in the evening, in the text, it reads, "the contribution from cooking sources to OOA was also important (Liu et al., 2018) as evidenced by the concurrent peaks of OOA with COA from their diurnal patterns."

Line 330: Maybe then this period was not appropriate to call it no heating?

Response: Compared to the "heating period" as we defined in late October, the period in early September (defined as "non-heating") featured much lower impacts from heating. Moreover, it is almost impossible to find an ideal "non-heating" period since sporadic use of solid fuels happen even in summer (Lin et al., 2019a). To be consistent with that at the kerbside, we tend to keep the term "non-heating" for the residential sampling site.

Line 333: The authors state that they chose the peat factor because it had a better correlation than BBOA reported by Ng et al. (2011a). But there is no other comparison to the rest of BBOA factors in the rest of literature. Recent studies has shown that the spectra of BBOA factors can vary a lot, so taking into account only one publication may lead to incorrect results and conclusions.

Response: We agree that BBOA profile vary a lot. In addition to the BBOA profile from the literature by Ng et al. (2011a), we have added the comparison with the wood burning OA profile as obtained in our previous fingerprinting experiments (Lin et al., 2017). We believe the profile from our own lab experiment (Lin et al., 2017) represents Irish cases better. It now reads, "The profile of the unconstrained primary OA factor is better correlated with the reference profile of peat burning OA ($R^2$=

0.58) than that of wood burning OA ($R^2$=0.38) from our previous study (Lin et al., 2017) and the averaged profile of biomass burning OA (BBOA; $R^2$=0.44) factor from the Ng et al. (2011a)."

Line 351: f44 is present and common to every OOA factor. What similarity want the authors to show?

Response: We want to show that OOA at the two sites has similar profile with $R^2$ of 0.95. It now reads, "The OOA mass spectral at the residential site featured high contribution at m/z 44 (i.e., *f44*) which was similar to the OOA ($R^2$=0.95 between the two OOA total mass spectral profile) at the kerbside".

Line 352: It has a pattern. It peaks at night. Suggestions?

Response: The diurnal cycle of OOA at the residential site showed only slightly elevated concentration in the evening (Fig.

6b) due to the processing/condensation of VOCs from e.g., residential heating. But compared to that at the kerbside, OOA at the residential site is more homogenously distributed across the data, suggesting most of the OOA at the residential site was likely due to regional transport. We have added such comments in the revised manuscript.

Line 397: What is the R2 between the time series of the factors? Does the correlation indicate common source of the factors?

(more factors than should be used etc?)

Response: R2 was in the range of 0.7-0.8 between the time series of the factors. The good correlation is explained by the co-emissions from heating sources (e.g., wood, peat burning) in the evening. Therefore, we took advantage of ME-2 to evaluate their respective contribution by constraining their reference profiles (Lin et al., 2017).

Line 412: This refer to Kerbside site?

Response: Yes. Now clarified.

Section 3.5. Discuss more. This is the main conclusion of this work and should include more information.

Response: We have now added more discussion in Sect 3.5. It now reads, "The HOA/BC ratio was ~0.17 as retrieved from the slope of the linear regression between HOA and BC with a coefficient of determination ($R^2$) being 0.61 for the non-heating period at the kerbside (Fig. S10). This points to, on average, 6 times higher traffic contribution to BC as compared to HOA. Note that HOA represented only the primary fraction of the organic aerosol emitted from the traffic. The secondary formation from the oxidation of precursor gaseous emissions from traffic was also important, as indicated by the elevated OOA concentration in the morning rush hours at the kerbside (Fig. 5b). This is consistent with the smog chamber studies by Gordon et al. (2014), which showed substantial amount of secondary OA formation from the precursor gases emitted by diesel vehicles with no aftertreatment techniques. In this study, the morning OOA peaks at the kerbside were mostly coinciding with the increase in HOA, pointing to a fast formation of secondary organic aerosol (i.e., from minutes to hours). However, the latter requires further investigations (e.g., real-time measurement of its precursor gases, oxidants etc.) to better understand the formation pathways in a real urban environment.

Regarding the HOA to BC ratio for the vehicular emissions, only the period of non-heating at the kerbside was considered. This is because HOA and BC were exclusively from traffic during this period. Note that the contribution from cooking to the ambient BC was considered negligible as there was a lack of BC peak during the lunch time at the kerbside (Fig. 4a). The negligible BC emissions from cooking were consistent with previous studies e.g., by Lee et al. (2015) and He et al. (2019). In contrast, during the heating period, HOA and BC had other sources (i.e., heating) in addition to traffic. For example, while both HOA and BC showed typical two rush hour peaks, an additional peak in the evening was observed during the heating period. Such evening peaks of HOA during the heating period could not be associated with vehicular emissions since, during the non-heating period, HOA showed only morning and afternoon rush-hour peaks. Also, the traffic pattern (i.e., with no increase in traffic in the evening) was not expected to change from the non-heating to the heating period. Instead, as discussed in Sect. 3.4, the evening peaks of HOA were associated with the use of oil for heating. Since the average diurnal profile of HOA, during the heating period, showed two rush hour peaks as during non-heating period and one additional evening heating peak, 67% of HOA was apportioned to be from vehicular emissions and 33% from the oil heating. As a comparison, at the residential site during the heating period, HOA was nearly exclusively (95%) from heating because it showed substantially higher evening peak than the morning rush hour peak.

The HOA/BC ratio in Dublin was lower than the HOA/BC ratios reported for the gasoline vehicles-dominated environment (0.9-1.7) but was within the range for the diesel environment (0.03-0.61) (DeWitt et al., 2015), indicating that most of the traffic emissions were from diesel vehicles. The large increment of HOA and BC at the kerbside in the Dublin city centre holds important implication for its adverse health impacts since the International Agency for Research on Cancer associated the exposure to the diesel engine exhaust with an increased rate of lung cancer. In Ireland, diesel fuel accounted for 73% of the on-road transport energy in 2017 (Fig. S11a). Figure S11a also shows an increasing trend of diesel fuel usage, pointing to a poorer air quality in the predictable future at the kerbside if diesel vehicular emissions were not controlled.

This study also shows that vehicular emissions appear to impact the air quality adjacent to the roads while, in contrast, solid fuel burning has a large geographic impact, affecting overall air quality at the kerbside and residential sites examined in this investigation. Such large geographic impact suggests significant climate effects from the residential emissions (Butt et al., 2016) which contain a higher fraction of OA than that from traffic. However, surprisingly, the census data shows only a few households (<5%) consuming these solid fuels with the majority households (~95%) using natural gas and electricity as the primary heating sources in 2016 (Fig. 11b). Therefore, if the emissions from this small fraction of households were controlled, good overall air quality and lower climate forcing can be expected."

References: Should all have the same format.

Response: We have double checked the format.

Figure 1: Please redo it. Rathimnes is not very visible with this grey colour. Also, stucking components is creating problems like during the period of 4/10 to 23/10 at Kerbside site where it seems like there is no other component but BC.

Response: We have redone Fig. 1. We have changed the grey color of $PM_{2.5}$ to light blue. For periods of 7/10 to 23/10, the

ACSM instrument was not operating for maintenance. We have clarified this in the caption for Fig. 1, it now reads, "The data gaps at the kerbside (e.g., from 7/10 to 23/10) were due to ACSM malfunction." We tend to keep the stacking style of Fig. 1, as the stacking of the components presents better comparison of $PM_1$ with $PM_{2.5}$.

Figure S3. Move it to main manuscript.

Response: Fig. S3 is now moved to the main manuscript (now as Fig. 3 in the revised manuscript)

Figure 2a. During Sunday seems like there is only BC. Fix it.

Response: ACSM was not sampling due to instrument malfunction. We have clarified this in the caption, it now reads, "At the kerbside, the ACSM data gaps (e.g., on 16/9) were due to ACSM malfunction."

Figure 2b. BC seems like contributing nothing. Fix it.

Response: We don't have data for this period. We have now clarified this in the caption. It now reads, "As a reference, the average concentration from the week later was included for the calculation of relative contribution to $PM_1$."

Figures 4, 5, 6, 7 (a). Fraction of what?

Response: Fraction of the total OA. Now clarified.

In overall the manuscript has been improved but still lacks in the section of scientific results. In introduction the authors state that: "In particular, the diurnal pattern of HOA shows largely enhanced concentration in the evening when compared to that during the morning rush hours, suggesting residential heating is the major source of HOA in suburban Dublin. However, the relative importance of traffic and heating to HOA in different urban settings (e.g., kerbside and residential) and different seasons (e.g., heating and non-heating) remain poorly understood." There is an effort in linking these statements to the current work but the authors should clarify better the relation between HOA and residential heating, something that is not very clear in the current version. The importance of HOA is not that high during the heating periods and maybe title should be reconsidered.

Response: We have now added more discussion to clarify the relation of HOA between traffic and residential heating (please also see the reply to the previous comment). Compared to HOA, the average Dublin traffic emitted 6 times more BC which was one of the major components of $PM_1$ at the kerbside even during the heating period. Therefore, we tend to keep the title of the manuscript as this project was mostly about traffic emission and its impact on the air quality at different settings (i.e., kerbside vs. residential) in Dublin.

In Sect. 3.5, it now reads, "The HOA/BC ratio was ~0.17 as retrieved from the slope of the linear regression between HOA and BC with a coefficient of determination ($R^2$) being 0.61 for the non-heating period at the kerbside (Fig. S10). This points to, on average, 6 times higher traffic contribution to BC as compared to HOA. Note that HOA represented only the primary fraction of the organic aerosol emitted from the traffic. The secondary formation from the oxidation of precursor gaseous emissions from traffic was also important, as indicated by the elevated OOA concentration in the morning rush hours at the kerbside (Fig. 5b). This is consistent with the smog chamber studies by Gordon et al. (2014), which showed substantial amount of secondary OA formation from the precursor gases emitted by diesel vehicles with no aftertreatment techniques. In this study, the morning OOA peaks at the kerbside were mostly coinciding with the increase in HOA, pointing to a fast formation of secondary organic aerosol (i.e., from minutes to hours). However, the latter requires further investigations (e.g., real-time measurement of its precursor gases, oxidants etc.) to better understand the formation pathways in a real urban environment.

Regarding the HOA to BC ratio for the vehicular emissions, only the period of non-heating at the kerbside was considered. This is because HOA and BC were exclusively from traffic during this period. Note that the contribution from cooking to the ambient BC was considered negligible as there was a lack of BC peak during the lunch time at the kerbside (Fig. 4a). The negligible BC emissions from cooking were consistent with previous studies e.g., by Lee et al. (2015) and He et al. (2019). In contrast, during the heating period, HOA and BC had other sources (i.e., heating) in addition to traffic. For example, while both HOA and BC showed typical two rush hour peaks, an additional peak in the evening was observed during the heating period. Such evening peaks of HOA during the heating period could not be associated with vehicular emissions since, during the non-heating period, HOA showed only morning and afternoon rush-hour peaks. Also, the traffic pattern (i.e., with no increase in traffic in the evening) was not expected to change from the non-heating to the heating period. Instead, as discussed in Sect. 3.4, the evening peaks of HOA were associated with the use of oil for heating. Since the average diurnal profile of HOA, during the heating period, showed two rush hour peaks as during non-heating period and one additional evening heating peak, 67% of HOA was apportioned to be from vehicular emissions and 33% from the oil heating. As a comparison, at the residential site during the heating period, HOA was nearly exclusively (95%) from heating because it showed substantially higher evening peak than the morning rush hour peak. "

[revised manuscript text omitted]